# Epistasis studies reveal redundancy among calcium-dependent protein kinases in motility and invasion of malaria parasites

Hanwei Fang[1], Ana Rita Gomes [2,6], Natacha Klages[1], Paco Pino[1], Bohumil Maco[1], Eloise M. Walker[3], Zenon A. Zenonos[2,7], Fiona Angrisano [4], Jake Baum[4], Christian Doerig[5], David A. Baker [3], Oliver Billker [8] & Mathieu Brochet [1]

In malaria parasites, evolution of parasitism has been linked to functional optimisation. Despite this optimisation, most members of a calcium-dependent protein kinase (CDPK) family show genetic redundancy during erythrocytic proliferation. To identify relationships between phospho-signalling pathways, we here screen 294 genetic interactions among protein kinases in *Plasmodium berghei*. This reveals a synthetic negative interaction between a hypomorphic allele of the protein kinase G (PKG) and CDPK4 to control erythrocyte invasion which is conserved in *P. falciparum*. CDPK4 becomes critical when PKG-dependent calcium signals are attenuated to phosphorylate proteins important for the stability of the inner membrane complex, which serves as an anchor for the acto-myosin motor required for motility and invasion. Finally, we show that multiple kinases functionally complement CDPK4 during erythrocytic proliferation and transmission to the mosquito. This study reveals how CDPKs are wired within a stage-transcending signalling network to control motility and host cell invasion in malaria parasites.

[1] Department of Microbiology and Molecular Medicine, Faculty of Medicine, University of Geneva, Geneva CH-1211, Switzerland. [2] Malaria Programme, Wellcome Trust Sanger Institute, Hinxton CB10 1SA, UK. [3] Faculty of Infectious and Tropical Diseases, London School of Hygiene &Tropical Medicine, Keppel Street, London WC1E 7HT, UK. [4] Department of Life Sciences, Imperial College London, London SW7 2AZ, UK. [5] Department of Microbiology, Infection & Immunity Program, Monash Biomedicine Discovery Institute, Monash University, Clayton, VIC 3800, Australia. [6] Present address: UMR 5290 Mivegec, University of Montpellier, Montpellier 34295, France. [7] Present address: Antibody Discovery and Protein Engineering, MedImmune, Cambridge CB21 6GH, UK. [8] The Laboratory for Molecular Infection Medicine Sweden (MIMS), Department of Molecular Biology, Umeå University, Umeå SE-901 87, Sweden. These authors contributed equally: Hanwei Fang, Ana Rita Gomes, Natacha Klages. Correspondence and requests for materials should be addressed to M.B. (email: Mathieu.Brochet@unige.ch)

Malaria is an infectious disease caused by mosquito-borne parasites of the genus *Plasmodium*. Protein kinases and phosphatases play important roles in regulating parasite development throughout their lifecycle[1]. Gene knockout screens have revealed dozens of kinases that could not be disrupted in the asexual blood stages of either the human parasite, *P. falciparum*[2], or *P. berghei*[3–5], a model parasite which infects rodents. Other protein kinase genes can be readily disrupted in asexual blood stages, although their protein or transcript is expressed during these stages. This raises the possibility that despite the high degree of functional optimisation in the *Plasmodium* genome[6], some kinase functions may be masked, for instance, by compensatory roles of structurally related enzymes from the same family. Candidates include members of the CDPK family, mitogen-activated protein (MAP) kinases or tyrosine kinase-like (TKL) enzymes[7,8], as illustrated by the overexpression of the second MAPK (Pfmap-2) present in the *P. falciparum* kinome in parasites lacking the Pfmap-1 kinase[8,9]. Functional roles of redundant genes can be revealed by genetic interaction studies where the combination of mutations in two or more genes generates unexpected phenotypes. Redundancy in signalling networks is common and has been demonstrated with great detail in yeast through genetic interaction screens[10]. In search of functional links between protein kinases during erythrocytic proliferation of *P. berghei*, we screened 294 double and triple mutants. This revealed an unexpected negative interaction between calcium-dependent protein kinase 4 (CDPK4), a known regulator of cell cycle progression during male sexual development but redundant in asexual stages[11,12], and modified alleles of protein kinase G (PKG), an essential kinase which controls key calcium signals across the lifecycle of malaria parasites[13].

During the erythrocytic phase of their lifecycle, parasites reproduce asexually producing merozoites that egress from the host red blood cell (RBC) and invade new RBCs within seconds. Regulation of egress and invasion of RBCs relies on multiple intracellular messengers, including calcium and cyclic guanosine monophosphate (cGMP), but the detailed architecture of the underpinning signalling networks remains poorly understood[14]. *Plasmodium* components of cGMP signalling are much more diverged from mammalian enzymes and PKG is the only cGMP effector characterised to date. Reverse genetic studies in *P. berghei* and *P. falciparum* have highlighted its essential role in parasite development[2,3]. As a result, functional studies have been performed using a chemical genetic approach whereby chemical inhibition of PKG was achieved by structurally distinct compounds, a pyrrole compound 1 (C1) and an imidazopyridine compound 2 (C2)[15–17]. Mutations of the threonine gatekeeper residue of PKG to a larger glutamine residue conferred reduced sensitivity to both inhibitors during egress and invasion, indicating that PKG is a primary target in these stages. Using this approach, we previously showed a major role for PKG in controlling calcium mobilisation from internal stores prior to RBC egress[13]. This PKG-dependent calcium signal is likely transduced by CDPK5, which is also required for secretion of micronemal proteins essential for egress and invasion in *P. falciparum*[18]. Another CDPK expressed in merozoites, CDPK1, was shown to be phosphorylated in a PKG-dependent manner and is involved in merozoite invasion[16].

PKG was further shown to regulate other calcium signals during mosquito transmission. Following ingestion by a mosquito, PKG mediates calcium mobilisation in the early steps of gametogenesis[13], after which CDPK4 and CDPK1 are required to complete gametogenesis[11,12,19–21]. A similar functional relationship between PKG and CDPK4 might be conserved for the invasion of hepatocytes by sporozoites where both kinases are important for efficient gliding motility[22]. PKG activity is also

necessary to maintain high cellular calcium levels in ookinetes[13] where calcium activates CDPK3, a kinase that supports efficient gliding and invasion of the epithelium of the mosquito midgut[23]. Interestingly, a gliding phenotype can be restored in the absence of both CDPK3 and the cGMP-specific phosphodiesterase delta (PDEδ). In this case, the lack of PDEδ over-activates PKG suggesting that, at least, one other unidentified calcium effector is involved in the regulation of ookinete motility[24]. This apparent re-wiring suggests that a network of calcium effectors responds to PKG-mediated calcium signals.

Here, we have identified that in merozoites and ookinetes, PKG-dependent calcium signals activate multiple CDPKs, including CDPK1 and CDPK4, which show specific and complementary functions to sustain efficient gliding motility and invasion. These functions are likely mediated by the phosphorylation of multiple proteins required for the formation and the stability of the inner membrane complex (IMC). We also show that in ookinetes, PKG-dependent calcium signals are additionally effected by CDPK3 to control secretion of micronemal proteins. Altogether, this indicates that distinct CDPKs decode PKG-dependent calcium signals to coordinate microneme secretion with acto-myosin motor activity.

## Results

**A genetic screen reveals an interaction between *cdpk4* and *pkg*.** To search for genetic interactions between *P. berghei* protein kinase genes, parasites from a panel of mutant clones lacking a specific kinase were negatively selected for loss of the selection marker and then transfected with a pool of barcoded gene knockout (KO) vectors to inactivate another kinase in the same background (Fig. 1a). The competitive growth rate of each mutant within the pool was measured during days 4–8 post infection by barcode sequencing[4]. For the background lines, we focussed on the CDPK family and on the two atypical MAP kinases (Supplementary Data 1 and Supplementary Fig. 1A–D). In preliminary experiments, we found that a double mutant of *map1* and *map2* showed normal asexual growth (Supplementary Fig. 1A), and the double KO mutant was included in the screen as a single recipient background to identify interactors of either gene. Due to the essential role for PKG in calcium mobilisation upstream of CDPKs, we also included the inhibitor-resistant PKG[T619Q]-3xHA line and its inhibitor-sensitive control, PKG-3xHA. The library of KO vectors was comprised of 37 targeting vectors for protein kinases and 6 characterised vectors targeting unrelated genes for use as references (Supplementary Data 1 and ref. [4]).

The screen examined 294 pairwise or 3-way combinations of mutant kinase alleles, the vast majority of which showed no evidence of interaction (Fig. 1b and Supplementary Data 2). Among these 294 combinations, 98 interaction tests involved one of the 14 vectors targeting genes likely essential in wild type. None of these became viable in any of the mutant backgrounds, i.e., there were no instances of strong positive interactions. No cases of synthetic lethality were observed. Most notably, no negative interactions among CDPKs were detected, as none of the pairwise deletions of CDPKs showed reduced growth beyond that of the single mutants (Supplementary Fig. 2). However, two negative interactions were both statistically significant and of sufficient magnitude to warrant further investigation (Fig. 1b). Both involved disruption of *cdpk4* in the presence of a modified allele of *pkg*, with the drug-resistant *pkg*[T619Q]-3xHA allele ($\varepsilon = -0.44$, $p$ value = 0.009, two-tailed $t$-test) imparting a more pronounced fitness cost to *cdpk4* deletion, when compared with the drug-sensitive *pkg*-3xHA background ($\varepsilon = -0.30$, $p$ value = 0.02, two-tailed $t$-test). These data suggest that either the epitope

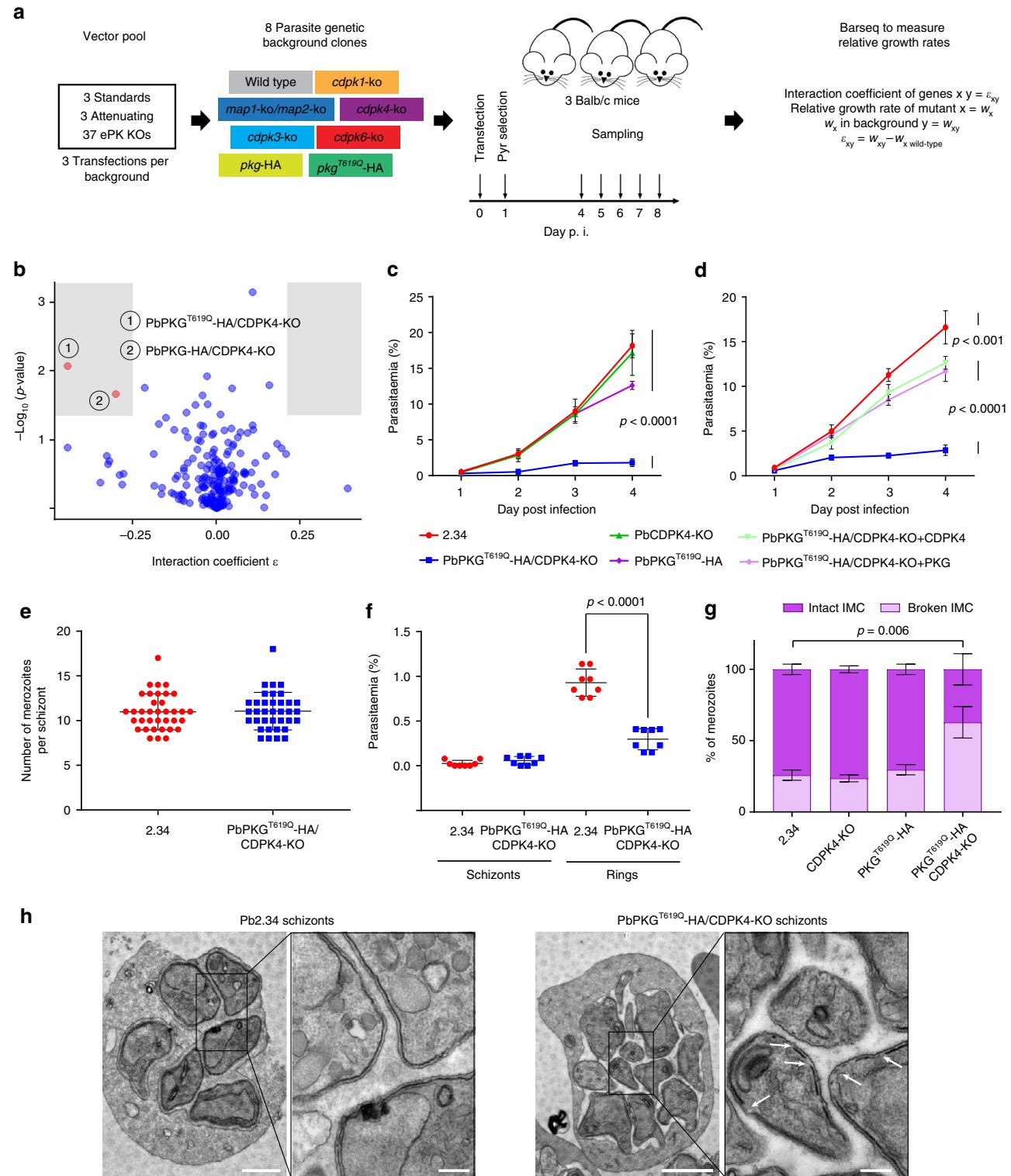

tag or the generic 3'-untranslated region (UTR) following the tag affected the function of PKG in a way that sensitised parasites to the deletion of *cdpk4*, an effect that was exacerbated by the T619Q substitution in the PKG active site.

To verify the negative interaction between *pkg* and *cdpk4*, a *cdpk4*-KO/*pkg*[T619Q]-3xHA clonal line was generated and found to have severely reduced growth (Fig. 1c) if compared to either of the single mutants (Supplementary Fig. 3A–B). Cis-complementing this line by re-inserting 3xHA epitope tagged

wild-type alleles of either *cdpk4* or *pkg* in their respective genomic loci (Supplementary Fig. 3C–D) restored parasite growth only partially (Supplementary Fig. 3E), possibly because the generic 3'UTR or the epitope tag used to monitor successful complementation creates hypomorphic alleles for both CDPK4 and PKG. Confirming this, cis-complementing the double mutant with non-epitope tagged wild-type alleles of either *cdpk4* or *pkg* (Supplementary Fig. 3F–G) restores erythrocytic growth almost to wild-type levels, validating the interaction between PKG and

**Fig. 1** A genetic interaction screen identifies a synthetic interaction between PbPKG and PbCDPK4 in asexual blood stages. **a** Schematic overview of the genetic interaction screen, definitions and analysis. **b** Interaction coefficients plotted against $p$ values. Interactions selected for validation (red) required $\varepsilon \leq -0.25$ or $\geq 0.25$ and $p$ value $< 0.05$ (shaded areas, two-tailed $t$-test). **c** Effect of $cdpk4$ deletion and $pkg$ mutagenesis on the growth of asexual blood stage parasites (error bars show standard deviations from the mean; 3 independent infections). **d** Complementation of $cdpk4$ gene deletion or $pkg$ mutagenesis with non-tagged wild-type alleles of $cdpk4$ or $pkg$ (error bars show standard deviations from the mean; 3 independent infections; two-way analysis of variance (ANOVA)). **e** Number of merozoites per schizont in control and PKG[T619Q]-3xHA parasites (data shown is from 3 independent in vitro cultures). **f** Parasitaemia observed 1 h after the intravenous injection of control or PKG[T619Q]-3xHA/CDPK4-KO mature segmented schizonts (data shown are from technical duplicates of four independent schizont cultures; error bars show standard deviations from the mean; two-tailed $t$-test). **g** Electron microscopy analysis of mature WT, CDPK4-KO, PKG[T619Q]-3xHA, and PKG[T619Q]-3xHA/CDPK4-KO schizonts (data from duplicates; error bars show standard deviations from the mean; two-way ANOVA, $n_{WT} = 140$, $n_{CDPK4-KO} = 103$, $n_{PKG T619Q -3xHA} = 48$, $n_{PKG T619Q -3xHA/CDPK4-KO} = 200$). **h** Representative electron microscopy pictures of mature WT and PKG[T619Q]-3xHA/CDPK4-KO schizonts. White arrows indicate gaps in the IMC that were more frequently observed in the transgenic line. Scale bar 1 μm (low magnification) and 200 nm (high magnification)

CDPK4 (Fig. 1d). A similar fitness cost of the generic 3'UTR and the epitope haemagglutinin (HA) tag is also observed during the sexual development of the parasite (Supplementary Note 1 and Supplementary Fig. 4A–G).

**The $pkg$/$cdpk4$ interaction is required for merozoite invasion.** Schizonts are mature erythrocytic stages that following asexual replication contain merozoites which, upon RBC egress, are able to infect new RBCs. Schizonts of the $cdpk4$-KO/$pkg$[T619Q]-3xHA line contain the same number of merozoites as the wild type (Fig. 1e), but when injected intravenously into mice, they show a reduced capacity to transform into ring stages while no significant amount of circulating schizonts is observed (Fig. 1f), suggesting that $pkg$ and $cdpk4$ genetically interact late in the growth cycle to control either the final stage of schizont maturation or invasion of new erythrocytes.

An ultrastructural analysis of $cdpk4$-KO/$pkg$[T619Q]-3xHA schizonts by transmission electron microscopy (TEM) revealed that a significantly higher number of mutant merozoites show a discontinuous IMC with gaps, while single mutants did not (Fig. 1g, h). The IMC is made up of alveoli beneath the plasma membrane and is important to maintain the stability of the parasite. Importantly, it is acting as an anchor for a protein complex known as the glideosome, which is crucial for host cell invasion.

**The $pkg$/$cdpk4$ interaction is conserved in *P. falciparum* merozoites.** To further investigate the interaction between PKG and CDPK4, we then focussed on egress and invasion in *P. falciparum*. We first aimed at generating a PfPKG[T618Q]/CDPK4-KO parasite line using CRISPR-Cas9. Attempts to disrupt the CDPK4 catalytic domain were only successful on a wild-type background (Supplementary Fig. 5A) and not in an existing line expressing the T618Q allele of PKG[25], suggesting that the observed interaction in *P. berghei* may translate to *P. falciparum*, but is likely synthetic lethal and thus not tractable by genetic deletion of $cdpk4$.

We reasoned, more conclusive evidence could be obtained with PKG inhibitors that may mimic the effect of the T618Q substitution in the CDPK4-KO background. First, we used the imidazopyridine C2, which was shown to inhibit PKG both in vitro and in vivo[25,26]. Synchronised mature schizonts were treated for 3 h with C2 and the number of ring and schizont parasites were then determined by flow cytometry (Supplementary Fig. 5B–C). A control line expressing the C2-resistant PKG[T618Q] allele is not affected at 4 μM, confirming PKG is the main target of C2 to block merozoite egress and invasion[26]. Unexpectedly, we observed a higher EC$_{50}$ of 0.8 μM for ring formation in the CDPK4-KO line compared with 0.4 μM for the 3D7 control line despite a similar half-maximal effective concentration (EC$_{50}$) for schizont rupture of 0.8 μM for both

lines (Fig. 2a). This indicates that for the same number of ruptured schizonts, the CDPK4-KO line gives rise to more ring parasites than the wild type in the presence of C2. This suggests that in the wild type, C2 has a second target important for invasion that is absent or not active in the CDPK4-KO line.

C2 has previously been suggested to inhibit both PfPKG and PfCDPK4 during male gametogenesis[25]. This dual specificity could be explained because both kinases share a relatively small gatekeeper residue (Thr and Ser, respectively) in the adenosine triphosphate (ATP) binding pocket required for inhibitor binding. The higher sensitivity to C2 of the 3D7 parental line for ring formation could therefore be due to a dual effect of C2 on both PKG and CDPK4. This possibility is supported by two lines of evidence. First, we confirm CDPK4 as an additional target for C2 in *P. falciparum* merozoites, by complementing the PfCDPK4-KO line with episomally expressed PbCDPK4-2xmyc or PbCDPK4[S147M]-2xmyc, respectively (Supplementary Fig. S5D). Complementation with PbCDPK4-2xmyc lowers ring formation in the presence of C2, while complementation with the PbCDPK4[S147M]-2xmyc allele does not (Fig. 2b). The S147M gatekeeper mutation in CDPK4 is known to confer resistance to another kinase inhibitor exploiting the small gatekeeper residue in the ATP binding pocket of CDPK4[12,19]. It is reasonable to assume that the same mutation would also block binding of C2, which in PKG is known to exploit the small gatekeeper of this kinase. Similarly, we observed that C2 also targets both PKG and CDPK4 during *P. berghei* gametogenesis, further substantiating the dual specificity of C2 across the malaria lifecycle (Supplementary Note 1 and Supplementary Fig. 4B–F). Secondly, we investigated whether an unrelated inhibitor of PKG that does not target CDPK4 (Supplementary Note 1, Supplementary Fig. 4C and ref. [27]), the imidazopyridazine Compound A, would show the same difference in EC$_{50}$ for ring formation as the CDPK4-KO line. As opposed to C2, Compound A shows the same EC$_{50}$ for schizont rupture and ring formation in both 3D7 and CDPK4-KO lines (Fig. 2c). This further points to subtle off-target effects on CDPK4 of C2 during merozoite invasion.

Altogether, these data indicate that C2 affects merozoite invasion by targeting both PKG and CDPK4 in the wild-type background, while in the CDPK4-KO line, the absence of CDPK4 is possibly compensated by one or multiple unknown C2-insensitive kinases, leading to a higher resistance to C2 for merozoite invasion.

**CDPK4 function is revealed by attenuated calcium signals.** Since in gametocytes CDPK4 is a known effector of a PKG-dependent calcium signal, we hypothesised that PfPKG[T618Q] represents a hypomorphic allele encoding a kinase that signals less effectively. Consistent with this idea, the T618Q mutation in recombinant PfPKG raises the affinity constant for ATP from 19.59 μM to 87.60 μM in vitro (Fig. 2d). In schizonts of the

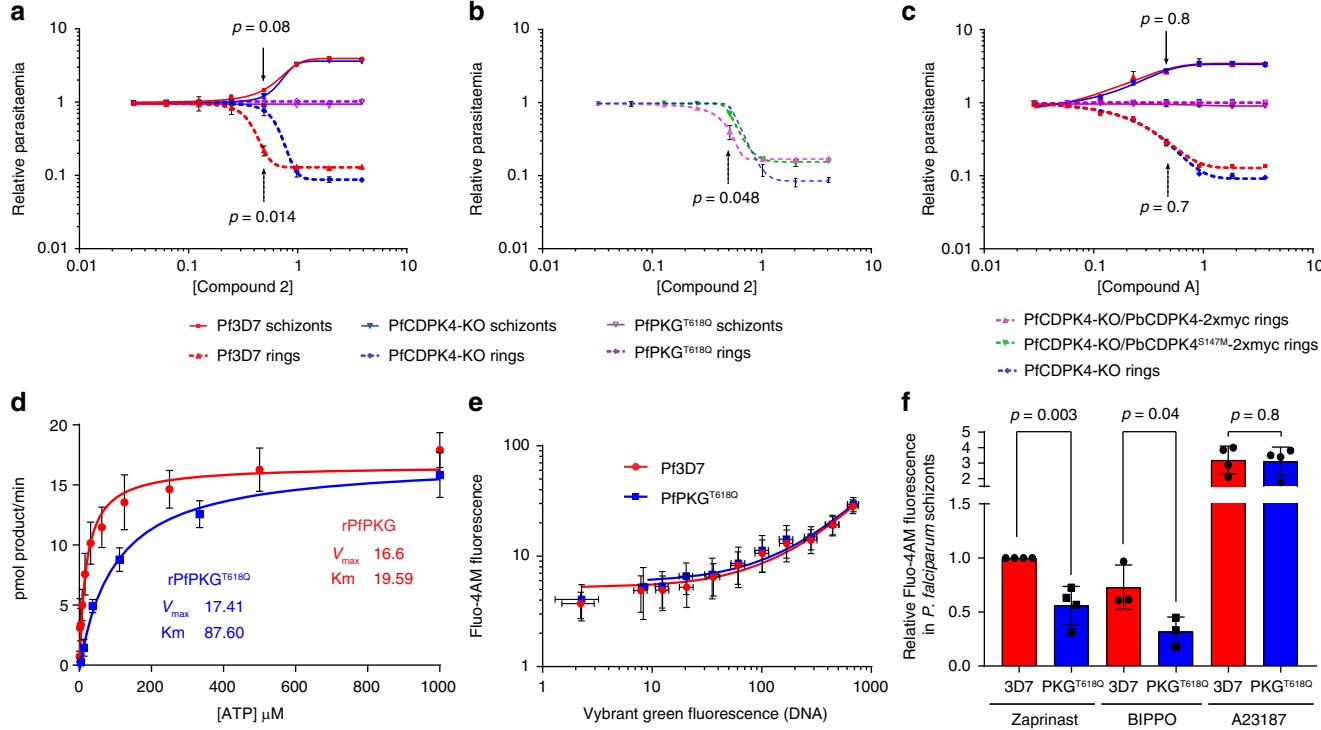

**Fig. 2** The CDPK4/PKG functional interaction is conserved to control *P. falciparum* merozoite invasion. **a** Difference in ring and schizont parasitaemia after 3 h of C2 treatment added on synchronised segmented schizonts in the control 3D7, PKG^T618Q, and CDPK4-KO lines (error bars show standard deviations from the mean; 2 independent cultures; two-tailed *t*-test). **b** Difference in ring and schizont parasitaemia after 3 h of C2 treatment added on synchronised segmented schizonts in the CDPK4-KO line and in daughter lines complemented with *P. berghei* CDPK4 or CDPK4^S147M alleles (error bars show standard deviations from the mean; 2 independent cultures; two-tailed *t*-test). **c** Difference in ring and schizont parasitaemia after 3 h of Compound A treatment added on synchronised segmented schizonts in the control 3D7, PKG^T618Q, and CDPK4-KO lines (error bars show standard deviations from the mean; 2 independent cultures; two-tailed *t*-test). **d** Determination of the ATP Km for recombinant PfPKG and PfPKG^T618Q. **e** Fluorescence of the cell permeable calcium probe Fluo-4-AM in non-synchronised 3D7 and PKG^T618Q parasites labelled with the DNA dye Vybrant Green (error bars show standard deviations from the mean; 2 biological replicates). **f** Relative fluorescence response of Pf3D7 and PfPKG^T618Q lines loaded with the calcium indicator Fluo-4-AM in response to the phosphodiesterase inhibitors Zaprinast and BIPPO as well as the ionophore A23187 (error bars show standard deviations from the mean; 3 or 4 biological replicates, two-tailed *t*-test)

PfPKG^T618Q line, this has no effect on baseline Ca^2+ levels in asexual blood stages (Fig. 2e). Raising cellular cGMP levels through phosphodiesterase inhibitors leads to a PKG-dependent increase in cytosolic Ca^2+ in schizonts[13]. This effect is significantly reduced in parasites expressing the PfPKG^T618Q allele (Fig. 2f and Supplementary Fig. 6A–C), although total cellular calcium available for release by an ionophore is unchanged. Similarly, we found that *P. berghei* parasites expressing the PKG^T619Q-3xHA allele show reduced intracellular Ca^2+ mobilisation during early gametogenesis (Supplementary Note 1 and Supplementary Fig. 4G). Altogether, this suggests that the increased requirement for CDPK4 in the PKG gatekeeper mutant lines results from weaker PKG-dependent calcium signals.

**CDPK4 is at the interface between the IMC and the glideosome.** While calcium levels can account for the functional interaction between *pkg* and *cdpk4* mutations, they do not explain how CDPK4 can affect RBC invasion by the merozoite. To address this question, we tagged CDPK4 in PbANKA 2.33, a line unable to produce gametocytes[28], and confirmed its expression in erythrocytic schizonts (Fig. 3a and Supplementary Fig. 7A). Immunofluorescence localisation shows a signal excluded from the nucleus with a slight enrichment at the merozoite periphery (Fig. 3a). In CDPK4-3xHA immunoprecipitates following crosslinking, 19 proteins are enriched over wild-type controls (Supplementary Data 3), including GAP40, MyoA and GAC, three

proteins essential for the IMC biogenesis or gliding motility[29,30]. In addition to MyoA, CDPK4 also immunoprecipitates an uncharacterised *Plasmodium*-specific myosin, MyoE (PBANKA_011220). Altogether, this further suggests that the redundant role of CDPK4 observed in invasion could be linked to the regulation of the molecular machinery that forms the IMC or provides the force for invasion.

Since we previously found GAP40^S448/S449 among a number of sites phosphorylated by an analogue-sensitive CDPK4^S147G (ref. [12]), we chose to first validate this putative interaction. Endogenously tagged GAP40 (Supplementary Fig. 7B) localises to the parasite periphery (Fig. 3a). Among the 20 proteins coimmunoprecipitated with GAP40-3xHA is CDPK4, in addition to known IMC and glideosome components (Fig. 3b and Supplementary Data 3). Since *gap40* is likely essential for asexual growth[12], we chose to examine specifically the relevance of GAP40^S448/S449 phosphorylation by substituting both serines with alanine residues. This does not affect parasite growth or integrity of the IMC in neither the wild-type nor the PKG^T619Q-3xHA/ CDPK4-KO background (Fig. 3c, d). The absence of an obvious phenotype for these substitutions could be explained either by the fact that phosphorylation of residues 448 and 449 of GAP40 is functionally not important or by the high levels of functional plasticity observed for the phosphorylation of glideosome components in apicomplexan parasites[31].

We were intrigued to find that GAP40-3xHA also coprecipitates another member of the CDPK family involved in

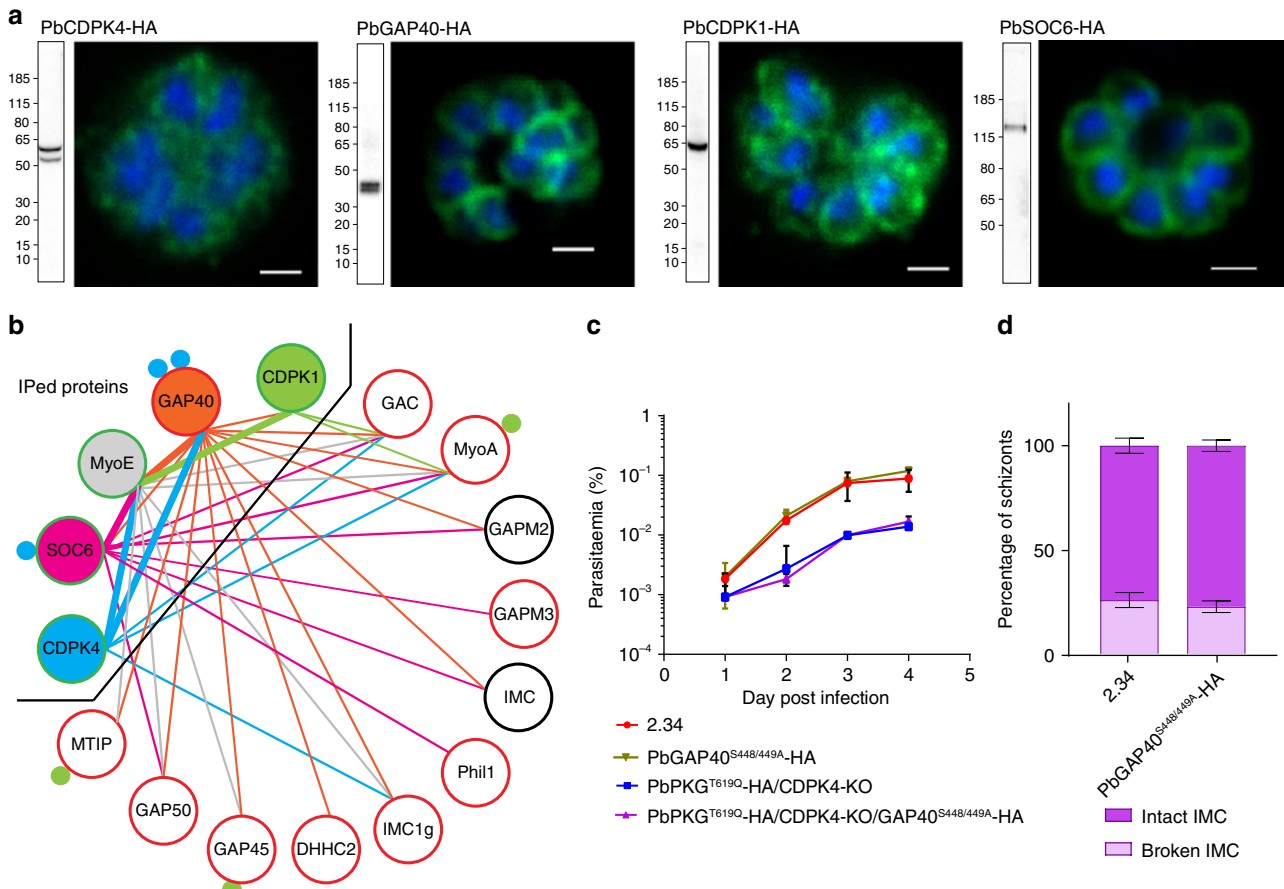

**Fig. 3** CDPK4 co-immunoprecipitates with components of the glideosome and the IMC. **a** Western blots and immunofluorescence analysis of schizonts expressing endogenously 3xHA tagged *cdpk4*, *gap40*, *cdpk1* and *soc6* alleles. Scale bars are 1 μm. **b** Protein interactions between IMC or glideosome proteins identified from GAP40-3xHA, SOC6-3xHA, MyoE-3xHA, CDPK4-3xHA and CDPK1-3xHA immunoprecipitates. Thick bars indicate that the interaction was identified from both immunoprecipitates. Blue and green filled circles denote the number of residues phosphorylated by CDPK4 ex vivo and by recombinant CDPK1 in vitro, respectively. The colour of the circle around the protein name indicates the requirement of its encoding gene for growth in asexual blood stages: red is essential, green is redundant; black denotes that the gene essentiality was not tested. **c** Effect of *gap40* mutagenesis on the growth of asexual blood stage parasites (error bars show standard deviations from 2 independent infections). **d** Electron microscopy analysis of mature WT and GAP40$^{S448/449A}$-3xHA schizonts (error bars show standard deviations from the mean; duplicates; two-tailed *t*-test; $n_{WT} = 140$, $n_{GAP40}S448/449A_{-3xHA} = 132$)

merozoite invasion, CDPK1, and again, MyoE. Epitope tagged CDPK1 and MyoE (Supplementary Fig. 7C–D) are enriched at the cell periphery (Fig. 3a and Supplementary Fig. 7D), and both proteins co-immunoprecipitate multiple glideosome or IMC components (Fig. 3b and Supplementary Data 3). Collectively, these results suggest MyoE may act as an alternative myosin of the motor complex and that both CDPK1 and CDPK4 are at the interface between the glideosome and the IMC. This may contribute to the genetic buffering observed when deleting these enzymes individually, although the initial screen found no strong evidence for a genetic interaction between them (Supplementary Data 1).

**CDPK4 phosphorylates a protein involved in IMC stability.** Immunoprecipitation of GAP40 or MyoE recovers multiple peptides from a protein of unknown function which, like GAP40 itself, was one of a small number of hits that emerged from our recent biochemical screen for substrates of CDPK4 (SOC proteins) in parasite lysates[12]. This protein, SOC6 (PBANKA_070770), has since been shown to interact with the IMC protein Phil1 in *P. berghei* schizonts[32] and with MyoA in *P. falciparum* schizonts[33], and may thus provide a molecular link

between CDPK4 and invasion. SOC6 is characterised by a C-terminal stretch of 106 amino acids that are relatively conserved across the syntenic orthologues of different malaria parasites (Supplementary Fig. 8A), but lacks obvious homologues in other apicomplexan genomes. SOC6 is further characterised by 4 to 15 tandem amino acid repeats that show sequence and position variability across species (Supplementary Fig. 8B). In *P. berghei*, the serine residue that CDPK4 phosphorylates in vitro lies in one such repeat of 54 amino acids (Supplementary Fig. 8A), which has prevented us from mutagenising specifically the phosphosite.

Endogenously tagged SOC6-3xHA (Supplementary Fig. 7E) localises to the cell periphery of merozoites (Fig. 3a) and immunoprecipitates peptides from multiple IMC, glideosome-associated proteins and glideosome proteins (Fig. 3b and Supplementary Data 3). Altogether, this indicates that SOC6 is also at the interface between the IMC and the glideosome. A SOC6-KO line shows a significant growth defect compared with wild type (Fig. 4a). While segmented SOC6-KO schizonts display the same number of merozoites as wild type (Fig. 4b), they show a reduced capacity to transform into ring stage parasites, while no accumulation of circulating SOC6-KO schizonts is observed

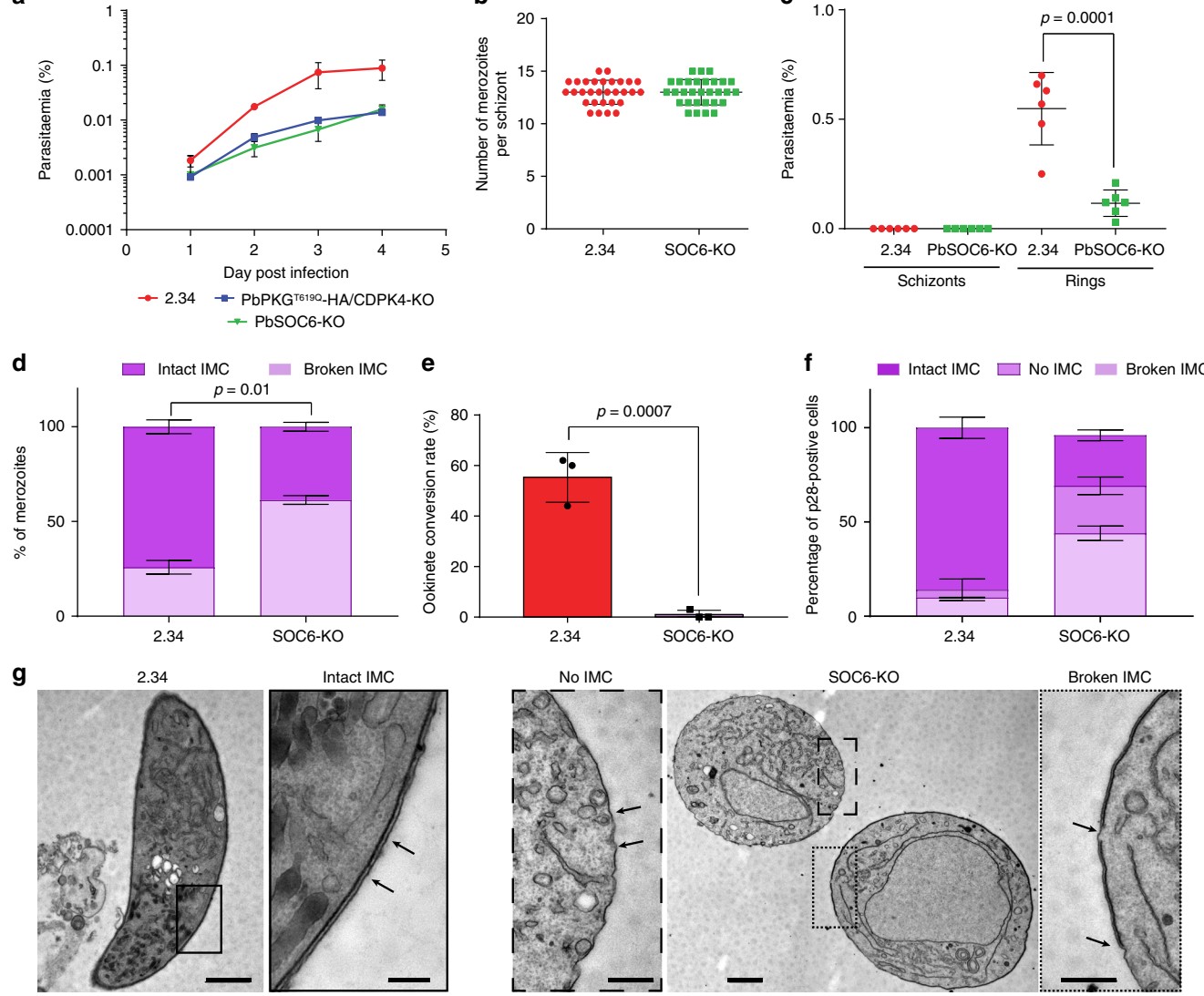

**Fig. 4** The CDPK4 substrate SOC6 is important for the IMC stability in merozoite and ookinete. **a** Effect of *soc6* deletion on the growth of asexual blood stage parasites (error bars show standard deviations from 2 independent infections). **b** Number of merozoites per schizont in control and SOC6-KO parasites (data shown is from 4 independent in vitro cultures). **c** Parasitaemia observed 1 h after the intravenous injection of control or SOC6-KO mature segmented schizonts (data shown is from four independent schizont cultures, two-tailed *t*-test). **d** Electron microscopy analysis of mature SOC6-KO and WT schizonts (error bars show standard deviations from duplicates, $n_{WT} = 140$, $n_{SOC6-KO} = 267$). **e** Ookinete conversion rate of 2.34 and SOC6-KO lines (error bars are standard deviations from three independent ookinete cultures, two-tailed *t*-test). **f** Ultrastructural analysis of WT and SOC6-KO highlighting the integrity of the IMC in WT and SOC6-KO p28-expressing cells (error bars are standard deviations from the mean, two ookinete cultures, $n_{WT} = 34$, $n_{SOC6-KO} = 27$). **g** Representative images of IMC defects observed in SOC6-KO parasites. Scale bar: 2.34, 1 μm (lower magnification) and 200 nm (higher magnification); SOC6-KO, 1 μm (lower magnification) and 500 nm (higher magnification)

(Fig. 4c). This indicates SOC6 is important either at the final stage of schizont maturation or to invade new RBCs. TEM of purified SOC6-KO schizonts reveals a discontinuous IMC as observed for PKG^T619Q-3xHA/CDPK4-KO transgenic (Fig. 4d and Supplementary Fig. 8C), suggesting that SOC6 is important for the formation or the stability of the IMC in merozoites.

To investigate the function of SOC6 further, we turned to the ookinete stage, which in *P. berghei* offers a tractable model to study the molecular motor that powers gliding motility. Ookinetes emerge from the zygote that forms after fertilisation of macrogametes by microgametes in the mosquito blood meal. Male gamete formation does not require SOC6[34], but the SOC6-KO nevertheless fails to form typical banana-shaped ookinetes (Fig. 4e). Again, TEM of SOC6-KO cells reveals either a

discontinuous IMC or the complete absence of an IMC below the plasma membrane (Fig. 4f, g), suggesting that SOC6 plays a conserved role to control the IMC formation or stability at multiple stages of the malaria lifecycle.

**CDPK4 and 1 functionally interact to control invasion and motility.** The presence of CDPK1 at the pellicle of *P. berghei* is consistent with its proposed function in phosphorylating a number of proteins possibly involved in motility and invasion[35,36]. Furthermore, there is evidence that CDPK1 functionally interacts with PKG in *P. falciparum* merozoites[37] and is important for erythrocyte invasion[38]. On the other hand, *P. falciparum* asexual blood stages can adapt to the loss of CDPK1[21],

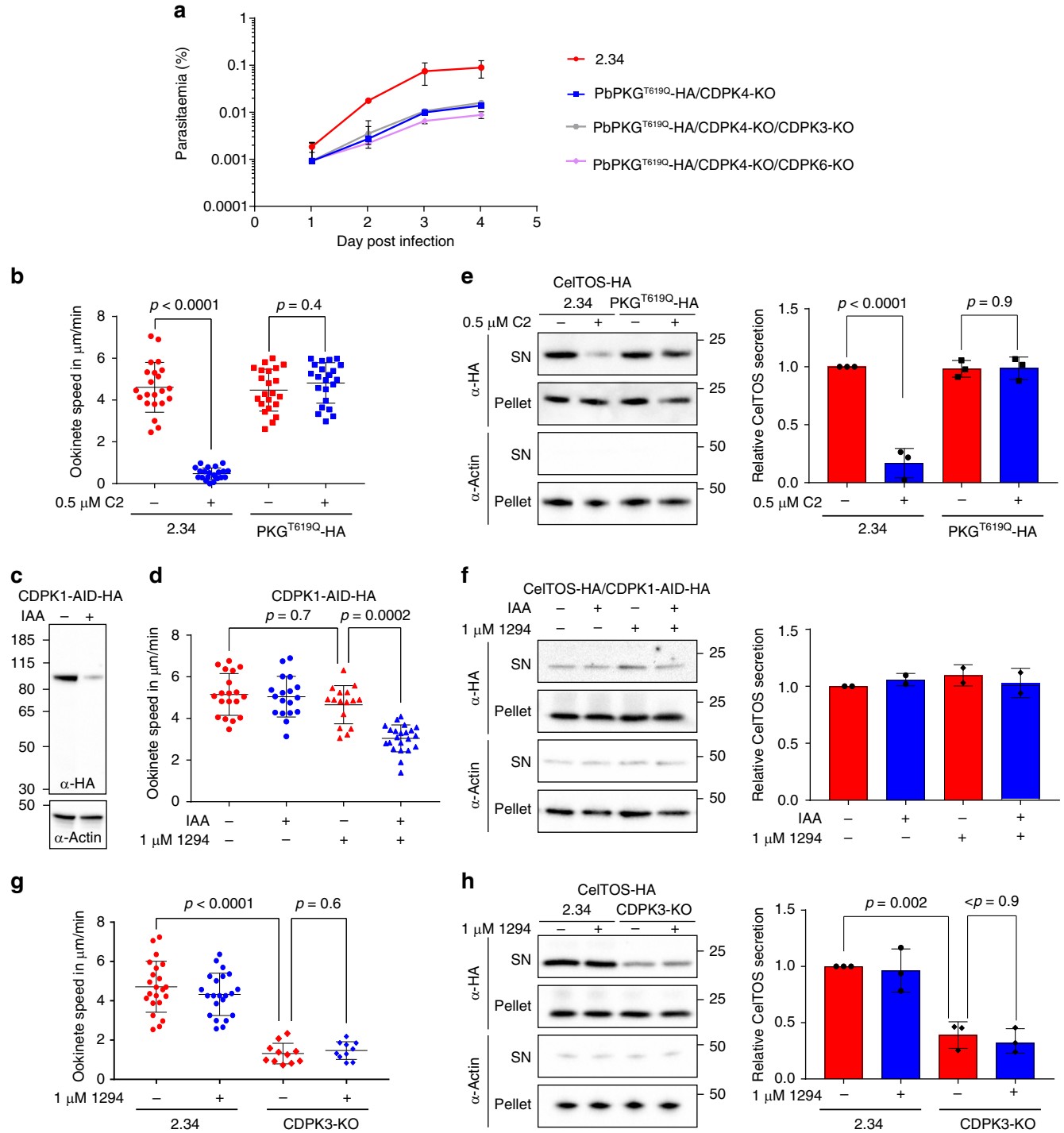

and in *P. berghei*, neither CDPK1-KO nor the double CDPK1-KO/CDPK4-KO nor PKG$^{T619Q}$-3xHA/CDPK1-KO parasites show a significant growth defect (Supplementary Data 1). However, we were not able to generate a triple PKG$^{T619Q}$-3xHA/CDPK4-KO/CDPK1-KO mutant, while CDPK3 and CDPK6 could be readily knocked out in the PKG$^{T619Q}$-3xHA/CDPK4-KO background (Fig. 5a). These results suggest that CDPK1 may partially complement the absence of CDPK4 and vice versa, and that another unidentified kinase is involved in this calcium signalling pathway downstream of PKG. In an attempt to identify other protein kinases that could compensate for loss of CDPK4 and CDPK1, we immunoprecipitated GAP40-3xHA or MyoE-

3xHA in the CDPK4-KO background, but no significant differences could be detected (Supplementary Fig. 7F).

PKG, CDPK4 and CDPK1 are abundantly expressed in ookinetes[20], and gliding requires PKG, since addition of 0.5 μM C2 almost completely blocks wild-type motility, while it has no effect on the C2-resistant PKG$^{T619Q}$-3xHA mutant (Fig. 5b). We next asked whether PKG-dependent gliding could be used to reveal roles for CDPK4 and CDPK1. As PbCDPK1 is essential for ookinetes to develop[20], we used a PbCDPK1-AID-HA line in which CDPK1 is inducibly degraded with the help of an auxin-dependent degron fused to the C terminus of the protein kinase (Fig. 5c and ref. [39]). PbCDPK1-AID-HA ookinetes show normal

**Fig. 5** PKG, CDPK4 and CDPK1 functionally interact to control asexual growth and ookinete motility. **a** Effect of *cdpk3* or *cdpk6* gene deletion in the PKG$^{T619Q}$-3xHA/CDPK4-KO background on the growth of asexual blood stage parasites (error bars show standard deviations from the mean; 2 independent infections). Note that a PKG$^{T619Q}$-3xHA/CDPK4-KO/CDPK1-KO line could not be generated. **b** Average gliding speed of 2.34 or PKG$^{T619Q}$-3xHA ookinetes ± 0.5 μM C2 (data show the average speed of single ookinetes from two independent biological replicates, two-way ANOVA). **c** Detection of CDPK1-AID-HA in ookinete lysate following a 30-min treatment ± auxin (IAA). **d** Average gliding speed of CDPK1-AID-HA ookinetes ± 1 μM 1294 and/or auxin (data show the average speed of single ookinetes from three independent biological replicates, two-way ANOVA). **e** Detection of the CelTOS-3xHA microneme protein and the cytosolic actin protein in the pellet and supernatant (SN) of 2.34 and PKG$^{T619Q}$-3xHA cultures ± 0.5 μM C2. The histogram shows quantification of the SN/pellet ratio for CelTOS-3xHA of each condition normalised to the SN/pellet ratio obtained in the 2.34 control in the absence of C2 (error bars show the standard deviations from the mean, three independent ookinete cultures, two-way ANOVA). **f** Detection of the CelTOS-3xHA microneme protein and the cytosolic actin protein in the pellet and supernatant of CDPK1-AID-HA cultures ± 1 μM 1294 and/or auxin. The histogram shows quantification of the SN/pellet ratio as in (**e**). **g** Average gliding speed of 2.34 and CDPK3-KO ookinetes ± 1 μM 1294 (data show the average speed of single ookinetes from the mean, two independent biological replicates, two-way ANOVA). **h** Detection of the CelTOS-3xHA microneme protein and the cytosolic actin protein in the pellet and supernatant of 2.34 or CDPK3-KO cultures ± 1 μM 1294. The histogram shows quantification of the SN/pellet ratio for CelTOS-3xHA as in (**e**)

gliding speed irrespective of auxin addition (Fig. 5d). Similarly, a CDPK4 selective inhibitor 1294[12,19] does not affect motility of PbCDPK1-AID-HA ookinetes at 1 μM in the absence of auxin. These data indicate that neither CDPK4 nor CDPK1 is individually required for gliding. However, when CDPK1-AID-HA is destabilised with auxin, the CDPK4 inhibitor significantly decreases ookinete speed, indicating that both enzymes control ookinete gliding and functionally complement each other (Fig. 5d).

In *P. falciparum* schizonts and *P. berghei* sporozoites, PKG controls microneme secretion[40], a process that is also critical to sustain gliding in ookinetes. C2 blocks the secretion of the ookinete microneme protein CelTOS-3xHA (Supplementary Fig. 9A) into the culture supernatant, specifically in the inhibitor-sensitive line (Fig. 5e), indicating that signalling through PKG is required for microneme secretion also in ookinetes. However, depletion of CDPK1 and chemical inhibition of CDPK4 does not affect secretion of CelTOS-3xHA either individually, or in combination (Fig. 5f). In marked contrast, deletion of CDPK3, an ookinete-specific CDPK needed for optimal gliding (Fig. 5g and refs. [23,41]), does reduce secretion of CelTOS-3xHA (Fig. 5h and Supplementary Fig. 9B, C). Complementation of *cdpk3* deletion ascertained that this effect was due to the absence of CDPK3 expression (Supplementary Fig. 9D-E). Furthermore, CDPK3 does not appear to interact functionally with CDPK4, since addition of 1294 does not decrease motility further in the CDPK3-KO (Fig. 5g). Altogether, this suggests that the main function of CDPK3 is to control microneme secretion downstream of PKG but independently of CDPK4, while CDPK1 and CDPK4 perform complementary functions in supporting efficient gliding.

## Discussion

Genetic interactions occur when mutations in two or more genes combine to generate an unexpected phenotype. Comprehensive interaction studies are a major undertaking owing to the combinatorial complexity of generating double mutants for each pairwise interaction tested, particularly in non-model organisms such as malaria parasites. A recent development in the genetic manipulation of *P. berghei* now enables large-scale reverse genetic studies in this species[4]. Using this approach, we describe the first genetic interaction screen in a malaria parasite. We examined 294 pairwise or 3-way combinations of mutant kinase alleles, the vast majority of which showed no evidence of interaction. We were only able to detect two negative interactions both involving disruption of the otherwise redundant *cdpk4* gene in the presence of hypomorphic alleles of *pkg*. This number appears to be low when compared to yeast, where around 1 million interactions were identified out of 23 million double mutants[42]. As signalling

cascades are more frequently associated with functional redundancy, as exemplified by the mammalian MAP kinases, nuclear factor-κB or Wnt pathways, these results suggest a possible reduction of genetic interaction networks in malaria parasites possibly linked to functional optimisation during evolution of parasitism[6]. However, our analysis mainly interrogated a subset of redundant kinases. As essential genes in yeast display five times as many interactions as non-essential genes[42], it is difficult to draw definitive conclusions regarding the extent of gene interaction networks in malaria parasites.

Genetic interaction networks highlight mechanistic connections between genes and their corresponding pathways. As PKG was known to regulate both merozoite egress and invasion, the negative interaction with CDPK4 strongly suggested a role for the latter in one of the two processes. Such a role for CDPK4 was unexpected as its known functions were to control unrelated processes during cell cycle transitions in microgametocytes[12]. Here, we reveal that CDPK4 is also important for the stability of the IMC, which serves as an anchor for the acto-myosin motor that is essential for *P. falciparum* invasion but not egress[43]. This functional interaction between PKG and CDPK4 to control gliding may be conserved in sporozoites where both enzymes support efficient gliding[22]. Signals transduced by CDPK4 are possibly mediated by phosphorylation of at least GAP40 and SOC6, which are both important for the biogenesis[44] and the stability of the IMC. SOC6 co-immunoprecipitated with Phil1 that was recently proposed to be important for IMC plate expansion[32] and it is possible that SOC6 plays a similar role. It is important to note that the molecular role of CDPK4-mediated phosphorylation remains unresolved as the SOC6-KO phenotype may be unrelated to its phosphorylation status. Further work will be necessary to comprehensively identify CDPK4 substrates in merozoites and dissect the molecular role of CDPK4-dependent phosphorylation events.

The impact of protein phosphorylation in controlling motility and invasion remains poorly understood in malaria parasites. A limited number of protein kinases have been shown to control gliding motility or invasion in *Plasmodium*. The strict requirement for PKG in both processes is probably pleiotropic due to its critical role in calcium regulation upstream of protein secretion and possibly regulation of the acto-myosin motor itself, as suggested by this study. The role of CDPKs downstream of PKG-dependent signals in motility and invasion also remains elusive. PfCDPK1 was proposed to directly regulate the glideosome as the recombinant enzyme phosphorylates PfMTIP and PfMyoA in vitro[35]. Phosphorylation of multiple IMC proteins was also shown to depend on CDPK1[38], further suggesting that CDPK1 participates in the assembly or stability of the IMC. The role of CDPK1 in invasion was also possibly linked to microneme

secretion. However, secretion of AMA1, a protein essential for merozoite invasion, was not CDPK1 dependent[38]. However, the exact contribution of CDPK1 to invasion remains unclear as merozoites lacking the kinase remain invasive[21,45] possibly due to the re-wiring of underlying signalling networks in its absence. Interestingly, CDPK5 was also shown to be essential for the secretion of micronemal proteins involved in merozoite invasion. This requirement was shown to become redundant when PKG is over-activated by a PDE inhibitor[18], suggesting that CDPK5 is part of the CDPK-dependent network downstream of PKG and may account for the redundancy of both CDPK4 and CDPK1 in merozoite and ookinetes. The role of CDPK3 in ookinete gliding was unknown and we show here that it controls microneme secretion. It is important to note that the essential protein kinase A (PKA) was proposed to be important for invasion in apicomplexan parasites by interplaying with cGMP- and calcium-dependent signalling[38,46]. Our genetic screen did not allow to study negative interactions with or among essential genes and it is highly likely that the cGMP/Ca$^{2+}$-dependent signalling network we describe here includes multiple other kinases including PKA. More work will be required to fully appreciate the exact architecture and plasticity of this signalling network. Nevertheless, this work suggests that in ookinetes and merozoites, PKG-dependent calcium signals are decoded by distinct but overlapping CDPK networks to coordinate microneme secretion and the acto-myosin motor with stage-transcending components, such as CDPK4, CDPK1, possibly CDPK5, and more stage-specific regulators such as CDPK3 (Fig. 6).

CDPKs are encoded by a large multigene family that is present in plants, protists, oomycetes and green algae, but is not found in animals and fungi[47]. It was proposed that protist and plant CDPK diversification into multiple gene family groups are independent of each other. Despite gene family expansion, CDPK gene sequences appear to be highly conserved, which could explain frequently observed functional redundancy among these enzymes[47]. For example, whereas biochemical approaches show distinct molecular functions of CDPKs in *Arabidopsis*, gene disruption phenotypes of CDPKs have only been rarely reported. This raises the intriguing question of why this gene family is amplified and diversified. It is clear that *Plasmodium* CDPKs evolved unique cellular functions[14]. However, our results indicate that they also retained functional redundancy, presumably to reach a threshold of global calcium-dependent kinase activity. It is interesting to note that CDPK1 did not functionally interact with hypomorphic alleles of PKG, suggesting it may require lower calcium levels than CDPK4. Interestingly, this CDPK network downstream of PKG-dependent calcium signals is conserved to transduce distinct signals in merozoites, gametocytes and ookinetes. These features may allow for a balance between evolvability to adapt to species- or stage-specific requirements and robustness of calcium-dependent signalling. For example, the requirement for CDPKs is slightly different between *P. berghei* and *P. falciparum* for gametogenesis[20,21] or merozoite invasion[21,37,45], suggesting that if the same network is involved in both processes, the wiring might be slightly different due to species-specific factors. On the other hand, requirement for the same kinases at unrelated developmental stages may also constrain the evolvability of CDPK networks. A similar situation is observed in the related apicomplexan parasite *Toxoplasma gondii*, where CDPKs show a high degree of redundancy[48] to control cellular processes including parasite egress and invasion. For example, microneme secretion universally depends on TgCDPK1, but only exhibits TgCDPK3 dependence when triggered by certain stimuli[49]. A better understanding of each kinase molecular function among different parasites and stages may further reveal how CDPK networks evolved for specific species or genus requirements within apicomplexan parasites.

By screening for genetic interactions among a subset of protein kinases, we have revealed an unexpected role for CDPK4 downstream of PKG-mediated calcium signals. To better map the signalling networks downstream of PKG-mediated signals across the lifecycle, it will be interesting to screen for interactions between CDPKs under conditions where PKG is over-activated, either by chemical inhibition or deletion of cGMP-specific phosphodiesterases. Similar analyses will also prove extremely valuable to gain a more comprehensive view of phospho-signalling pathways by, for example, screening for genetic interaction among kinases and phosphatases or using conditional approaches to study genetic interactions with essential genes.

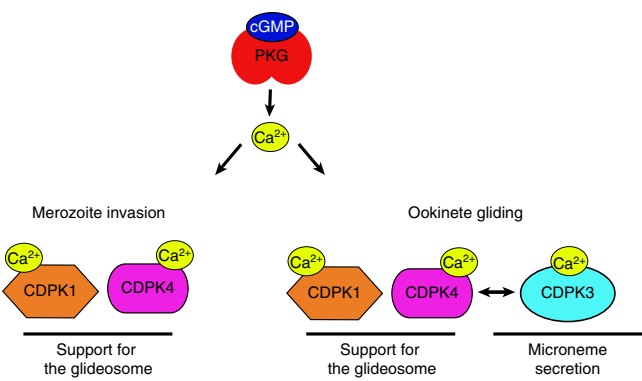

**Fig. 6** Schematic representation of the CDPK networks downstream of PKG-mediated calcium signals. PKG acts as a stage-transcending calcium regulator that activates multiple CDPKs with stage-specific functions. In merozoites, calcium activates both CDPK4 and 1 to support the activity of the acto-myosin motor. In ookinetes, PKG-dependent calcium signals are required for efficient gliding through CDPK4 and 1 activation and microneme secretion via CDPK3 regulation. Under physiological activation of PKG, CDPK1 and CDPK4 may functionally complement each other to sustain the acto-myosin motor function, while hyperactivation of PKG by chemical or genetic inhibition of phosphodiesterase or adaptation to gene deletion may allow further complementation by CDPK3 in ookinetes[24]. Note that other kinases such as CDPK5 or the protein kinase A (PKA) are possibly part of this network but the exact links with PKG and these kinases could not be revealed by this study and remain unmapped

## Methods

**Preparation of targeting vectors.** 3xHA tagging, knockout and allelic replacement constructs in *P. berghei* were generated by phage recombineering in *Escherichia coli* tryptic soy agar (TSA) bacterial strain with *Plasmo*GEM vectors (http://plasmogem.sanger.ac.uk/). Vectors available in the *Plasmo*GEM repository are listed in Supplementary Data 1. For final targeting vectors not available in the PlasmoGEM repository, generation tagging constructs was performed using sequential recombineering and gateway steps[50,51]. A list of oligonucleotides used in this study is available in Supplementary Data 4. For each gene of interest (goi), the Zeocin-resistance/Phe-sensitivity cassette was introduced using oligonucleotides *goi* HA-F x *goi* HA-R for 3xHA tagging. Substitution of the GAP40$^{S148/149A}$ residue was introduced using primer gap40S148 HA-F instead of gap40 HA-F. Mutations were confirmed by sequencing with primers *gap40*-QCR1 and GW1. The modified library inserts were released from the plasmid backbone using *Not*I.

To generate a new gateway entry cassette allowing simultaneously to introduce an in-frame AID-HA tag and express the Tir1 protein in ookinetes (generation of a CDPK3-KO/CelTOS-AID-HA/Tir1 line), the gateway entry vector for AID-HA tagging from the PlasmoGEM resource was first linearised by *Kpn*I. The *osTIR1* gene, *hsp70* promoter and *p28* 3'UTR were amplified from genomic DNA of *P. berghei* CDPK1-AID-HA line[39] with oligonucleotides ostir1 F x ostir1 R, Phsp70 F x Phsp70 R and p28 3'UTR F x p28 3'UTR R. Linearised vectors and PCR amplicons were then assembled by Gibson Assembly to create the circular GW-AID-HA-Tir1 vector.

For the PfCDPK4-KO construct, homology regions were cloned into the pcc1 plasmid. A first homology region mapping upstream of *pfcdpk4* was amplified with

oligos PfCDPK4 HR1-F and PfCDPK4 HR1-R and ligated into SacII/AflII digested pcc1. A second homology region mapping downstream of the region encoding CDPK4 catalytic domain was amplified using PfCDPK4 HR2-F and PfCDPK4 HR2-R and ligated into NcoI/AvrII digested pcc1 to generate pcc1-CDPK4-KO plasmid. The gRNA oligonucleotides PfCDPK4 gRNA F and PfCDPK4 gRNA R were ligated into the BstGZI digested pUF1-Cas9 plasmid. To generate a PfPKG[T618Q]/CDPK4-KO, we swapped the hdhfr gene of the pcc1-CDPK4-KO plasmid by digesting the vector with AflII and HindIII. The bsd gene was amplified using BSDswap F x BSDswap R. The resulting amplicon was digested with AflII and HindIII and ligated into the above vector. Sequence of the bsd gene was confirmed by Sanger sequencing using primers bsd seq1 and 2.

For the PfCDPK4-KO complementation constructs, the hDHFR selection cassette of plasmid pSD152 was replaced by a BSD selection cassette from pminiBSD2 by restriction ligation using PstI and KpnI resulting in pSD153. To introduce the S147M substation, pSD153 was digested with NheI and BspI. A replacement insert containing the substitution was amplified with primers cdpk4-S147M F and cdpk4-S147M R and further Gibson assembled in the digested plasmid. The sequence of the cdpk4 gene was confirmed by Sanger sequencing using primers cdpk4-seq1F to cdpk4-seq4F and GW1.

**Parasites and animals**. P. berghei strain ANKA[52] derived clone 2.34[11] and derived transgenic lines were maintained in female CD1 outbred mice. The parasitaemia of infected animals was determined by methanol-fixed and Giemsa-stained thin blood smears. CD1 outbred mice were obtained from Charles River Laboratories. Mice were specific-pathogen free and subjected to regular pathogen monitoring by sentinel screening. They were housed in individually ventilated cages furnished with a cardboard mouse house and Nestlet. Mice were maintained at $21 \pm 2\,^{\circ}\text{C}$ under a 12 h light/dark cycle and given commercially prepared autoclaved dry rodent diet and water ad libitum. Female mice were used for experimentation at 6–10 weeks of age and were randomly selected for parasite infection. The investigator was blinded to the parasite group allocation for the intravenous injection of P. berghei schizonts. Animal experiments were conducted with the authorisation numbers GE/82/15 and GE/201/17 according to the guidelines and regulations issued by the Swiss Federal Veterinary Office or under a license from the UK Home Office in accordance with national and European animal welfare guidelines. For the analysis of isolated transgenic parasites, the sample size was chosen to ensure a power of at least 80% using G*Power (http://www.gpower.hhu.de/). Statistical analyses were performed using GraphPad Prism 7.

**P. berghei culture and transfection**. Schizonts for transfection were purified from overnight cultures on a Histodenz cushion made up from 55% of a Histodenz stock and 45% phosphate-buffered saline (PBS). Purified parasites were harvested from the interphase and centrifuged at $500 \times g$ for 3 min. Two protocols have been used for parasite electroporation. In the first one used at the University of Geneva, cells were re-suspended in 100 μL Amaxa Basic parasite Nucleofector solution (Lonza), added to 10–20 μg of precipitated DNA re-suspended in 10 μL of $H_2O$ and electroporated using the U-033 program of the Amaxa Nucleofector II. Transfected parasites were resupended in 200 μL of fresh red blood cells and injected intraperitoneally into mice. Selection with 0.14 mg/mL pyrimethamine (Sigma) in drinking water (pH ~4.5) was initiated from day 1 post infection. In the second protocol used at the Sanger Institute, schizonts were re-suspended in 18 μL Amaxa Primary cells Nucleofector solution (Lonza) and added to ~5 μg of precipitated DNA re-suspended in 5 μL of $H_2O$. Cells were electroporated using the FI-115 program of the Amaxa Nucleofector 4D and transfected parasites were directly injected intravenously into the tail vein of mice. Selection with 0.07 mg/mL pyrimethamine (Sigma) in drinking water (pH ~4.5) was initiated from day 1 post infection. Negative selection of parasites expressing yFCU (a bifunctional protein that combines yeast cytosine deaminase and uridyl phosphoribosyl transferase) was performed through the administration of 5 fluorocytosine (1 mg/mL, Sigma) via the drinking water[53]. Each mutant was genotyped using different combinations of primers, specific for either the WT or modified locus on both sides of the targeted locus (experimental designs are shown in Supplementary Figures). For allelic replacements, sequences were confirmed by Sanger sequencing using indicated primers. WT DNA controls were included in each genotyping panel.

For ookinete cultures, parasites were maintained in phenyl hydrazine-treated mice. Ookinetes were produced in vitro by adding 1 volume of high gametocytaemia blood in 30 volumes of ookinete medium (RPMI-1640 containing 25 mM HEPES, 10% foetal calf serum, 100 μM xanthurenic acid, pH 7.5) and incubated at 19 °C for 18–24 h. Ookinete conversion efficiency was determined by live staining of ookinetes and activated macrogametes with Cy3-conjugated 13.1 monoclonal antibody against p28. The conversion rate was determined as the number of banana-shaped ookinetes as a percentage of the total number of Cy3-fluorescent cells. For secretion assays, ookinetes were purified using paramagnetic anti-mouse IgG beads (Life Technologies) coated with anti-p28 mouse monoclonal antibody (13.1). Purified ookinete were then incubated for 1 h at 19 °C and the supernatant was separated from the parasite pellet by centrifugation ($500 \times g$, 3 min). For motility assays, ookinetes in culture medium were added to an equal volume of Matrigel (BD Bioscience) containing dimethyl sulphoxide (DMSO) or inhibitors kept on ice, mixed thoroughly, dropped onto a slide, covered with a

cover slip and sealed with nail polish. After randomly identifying a field containing ookinetes, time-lapse videos were taken (1 frame every 10 s, for 10 min) on a Nikon eclipse Ty with a 100× objective at room temperature. Movies were analysed with Fiji and the Manual Tracking plugin (http://pacific.mpi-cbg.de/wiki/index.php/Manual_Tracking).

**P. falciparum culture and transfection**. P. falciparum strain 3D7 and derivatives were grown in erythrocytes in RPMI-1640 medium with glutamine (Gibco), 0.2% sodium bicarbonate, 25 mM HEPES, 0.2% glucose, 5% human serum and 0.1% Albumax II (Life Technologies). Parasite cultures were kept synchronised by double sorbitol treatments. Late-stage parasites were purified from highly synchronous cultures using Percoll (GE Healthcare) and used for invasion assays and transfection.

For P. falciparum transfections purified mature schizonts were electroporated with 40 μg of circular plasmid DNA using the Amaxa Nucleofactor II (Lonza) and the Nucleofector™ Kits for Parasites (Lonza). Selection for transgenic parasites was performed by culture in medium containing 2.5 nM WR99210 (Jacobus Pharmaceuticals) or 2 nM blastidicin. When genome integration was detected by diagnostic PCR, parasites were cloned by limiting dilution[54].

For egress/invasion assays, highly synchronous ring stages were plated in triplicate at 1–3% parasitaemia and 1% haematocrit. Cultures were treated with various doses of inhibitors around 2 h prior to egress and allowed to mature until schizonts stage, egress and re-invade for 1 h. Parasites were labelled with the DNA dye Vybrant dye cycle Green (life Technologies) for 30 min and analysed using a Beckman Coulter Gallios 4. Per sample, >50,000 cells were analysed with Kaluza Analysis and all measurements were performed on two independent biological replicates, at least.

**Genetic interaction screen**. A set of 43 KO vectors were pooled into 8 identical pools, each containing 100 ng of each vector. Each pool included spike-in DNA of control vectors, and was prepared and transfected in triplicates into P. berghei schizonts[4]. Identification numbers for all vectors included in this study are shown in Supplementary Data 1 and can be used to access details of each vector design on the PlasmoGEM database (http://plasmogem.sanger.ac.uk). Infections were sampled at the same time of each day between days 4 and 8 post infection. Parasite genomic DNA was phenol/chloroform extracted from these samples[4]. Growth rates of single, double and triple mutants were measured using the gene-specific 11 bp barcodes present in each of PlasmoGEM vectors, which uniquely label the parasite's genome upon vector integration. Barcodes were amplified linearly from genomic DNA extracts and counted on an Illumina MiSeq. For that, amplicon-based Illumina sequencing libraries were prepared using a nested PCR approach to yield 234 bp long amplicons containing sample-specific indexes that were pooled equimolarly in groups of 32. Growth rates and relative fitness of each single, double or triple mutants were calculated using an algorithm developed in ref. [6]. These data were used to calculate interaction coefficients using a subtractive model whereby the relative fitness of each single mutant (i.e., transfection performed on the wild-type background) was subtracted from the relative fitness measured for each pairwise or three-way test (i.e., transfections performed on the different mutant backgrounds): Relative fitness of mutant x in wild-type background = $w_{x0}$, $w_x$ in background y = $w_{xy}$, interaction coefficient genes xy = $\varepsilon_{xy}$ where $\varepsilon_{xy} = w_{xy} - w_{x0}$.

**Protein analysis**. Figure source data for all western blots are shown in Supplementary Fig. 10. Co-immunoprecipitation of CDPK4-3xHA, CDPK1-3xHA, MyoE-3xHA, SOC6-3xHA, GAP40-3xHA protein complexes were performed with schizont fixed for 10 min with 1% formaldehyde, lysed in RIPA buffer and the supernatant was subjected to affinity purification with magnetics beads conjugated with monoclonal anti-HA antibody, clone 3F10 (Sigma-Aldrich, reference 000000011867431001). A WT control was included in parallel and proteins for which we recovered peptides in the WT control were not retained for further analysis. Magnetic beads were suspended in 100 μL of 6 M Urea in 50 mM ammonium bicarbonate (AB). To this solution, 2 μl of dithiothreitol (DTT; 50 mM in liquid chromatography–mass spectrometry (LC-MS) grade water) were added and the reduction was carried out at 37 °C for 1 h. Alkylation was performed by adding 2 μL of iodoacetamide (400 mM in distilled water) for 1 h at room temperature in the dark. Urea concentration was lowered to 1 M with 50 mM AB, and protein digestion was performed overnight at 37 °C with 15 μL of freshly prepared trypsin Promega (0.2 μg/μL in AB). After bead removal, the sample was desalted with a C18 microspin column (Harvard Apparatus, Holliston, MA, USA), dried under speed-vacuum, and re-dissolved in $H_2O/CH_3CN/FA$ 94.9/5/0.1 before liquid chromatography–electrospray ionisation–tandem mass chromatography (LC-ESI-MS/MS) analysis. LC-ESI-MS/MS was performed on a Q-Exactive Hybrid Quadrupole-Orbitrap Mass Spectrometer (Thermo Fisher Scientific) equipped with an Easy nLC 1000 system (Thermo Fisher Scientific). Peptides were trapped on an Acclaim pepmap100, C18, 3 μm, 75 μm x 20 mm nano trap-column (Thermo Fisher Scientific) and separated on a 75 μm x 500 mm, C18, 2 μm Easy-Spray column (Thermo Fisher Scientific). The analytical separation was run for 90 min using a gradient of $H_2O/FA$ 99.9%/0.1% (solvent A) and $CH_3CN/FA$ 99.9%/0.1% (solvent B). The gradient was run as follows: 0–5 min 95% A and 5% B, then to 65% A and

35% B in 60 min, then to 10% A and 90% B in 10 min, and finally stay at 10% A and 90% B for 15 min. The entire run was at a flow rate of 250 nL/min. ESI was performed in positive mode. For MS survey scans, the resolution was set to 70,000, the ion population was set to $3 \times 10^6$ with a maximum injection time of 100 ms and a scan range window from 400 to 2000 $m/z$. For MS2 data-dependent acquisition, up to 15 precursor ions were selected for higher-energy collisional dissociation. The resolution was set to 17,500, the ion population was set to $1 \times 10^5$ with a maximum injection time of 50 ms and an isolation width of 1.6 $m/z$ units. The normalised collision energies were set to 27%. Peak lists were generated from raw data using the MS Convert conversion tool from ProteoWizard. The peaklist files were searched against the *Plasmodium berghei*_ANKA database (*Plasmodium* Genomic Resource, release 28, 5076 entries) using Mascot search engine (Matrix Science, London, UK; version 2.5.1). Trypsin was selected as the enzyme, with one potential missed cleavage. Fragment ion mass tolerance was set to 0.020 Da and parent ion tolerance to 10.0 ppm. Variable amino acid modification was oxidised methionine and fixed amino acid modification was carbamidomethyl cysteine. The mascot search was validated using Scaffold 4.7.3 (Proteome Software Inc., Portland, OR). Protein identifications were accepted if they could be established at greater than 95.0% probability and contained at least 3 identified peptides.

**Transmission electron microscopy**. RBCs infected with *P. berghei* or extracellular ookinetes were fixed with 2.5% glutaraldehyde (Electron Microscopy Sciences) and 2.0% paraformaldehyde (Electron Microscopy Sciences) in 10 mM PBS pH 7.4 for 1 h at room temperature. Pelleted cells were embedded in 3% low melted agarose (Eurobio) in 10 mM PBS and dissected in small pieces for easier handling and to prevent loss of cells during subsequent processing steps. After extensive washing ($5 \times 5$ min) in 0.1 M sodium cacodylate buffer pH 7.4 (Sigma), samples were post-fixed with 1% osmium tetroxide (Electron Microscopy Sciences) reduced with 1.5% ferrocyanide (Sigma) in 0.1 M sodium cacodylate buffer pH 7.4 for 1 h at room temperature, followed by post-fixation with 1% osmium tetroxide alone (Electron Microscopy Sciences) in 0.1 M sodium cacodylate buffer pH 7.4 for 1 h at room temperature. After washing with double distilled water ($2 \times 5$ min) samples were en-block post-stained with 1% aqueous uranyl acetate (Electron Microscopy Sciences) for 1 h at room temperature. Samples were then washed with double distilled water for 5 min and dehydrated in graded ethanol series ($2 \times 50$%, $1 \times 70$%, $1 \times 90$%, $1 \times 95$%, and $2 \times 100$% for 3 min each wash). Samples were then infiltrated at room temperature with Durcupan resin (Sigma) mixed 100% ethanol at 1:2, 1:1 and 2:1 for 30 min each step, followed by fresh pure Durcupan resin for $2 \times 30$ min and transferred into fresh pure Durcupan resin for 2 h. Finally, samples were embedded in fresh Durcupan resin-filled small thin-wall PCR tubes and polymerised and cured at 60 °C for 24 h. Ultrathin sections (60 nm) were cut with Leica Ultracut UCT microtome (Leica Microsystems) and diamond knife (DiATOME) and collected onto 2 mm single slot copper grids (Electron Microscopy Sciences) coated with 1% Pioloform plastic support film. Sections were then examined and TEM images collected using Tecnai 20 TEM (FEI) electron microscope operating at 80 kV and equipped with a side-mounted MegaView III CCD camera (Olympus Soft-Imaging Systems) controlled by iTEM acquisition software (Olympus Soft-Imaging Systems).

**Calcium measurements in *P. falciparum* schizonts**. Changes in the levels of intracellular free calcium were measured using Fluo-4 (Sigma) loaded *P. falciparum* late-stage schizonts. Excitation was measured using a SPECTRAmax microplate fluorometer. Fluorescence levels were compared to a baseline read prior to the addition of a test reagent. Schizonts were purified magnetically (Macs; Milteny Biotec) and pelleted by centrifugation for 2 min at $500 \times g$. Parasites were re-suspended in $10 \times$ warm Ringer Buffer (122.5 mM NaCl, 5.4 mM KCl, 0.8 mM MgCl$_2$, 11 mM HEPES, 10 mM D-Glucose, 1 mM NaH$_2$PO$_4$) to $1$–$2 \times 10^8$ parasites/mL and 2 µL of 5 mM Fluo-4 was added to 1 ml of parasite preparation. Cells were incubated with Fluo-4 at 37 °C for 45 min and washed twice in warm Ringer buffer and incubated for 20 min for de-esterification followed by a further two washes. The pellet was re-suspended in Ringer buffer and plated out on the bottom half of a 96 well plate. The excitation of the cells was measured at 20 s intervals for a period of 3 min to achieve a baseline read. Compounds were then added onto the cells and reads were recorded for a further 5 min.

**Expression and purification of recombinant PKG**. Full-length PfPKG with native codon usage was cloned into the pTrcHisC plasmid (Life Technologies) that includes an N-terminal His-tag as described in ref. [55]. PfPKG with threonine 618 replaced with a glutamine (PfPKG$^{T618Q}$) was cloned into the same plasmid[25]. Recombinant proteins were generated and purified using a protocol based on that described in ref. [55]. Briefly, freshly transformed *E. coli* Rosetta2 (DE3) were used for expression of recombinant PfPKG. The 500 mL cultures in LB Rich Broth (containing 50 µg/mL carbenicillin and 34 µg/mL chloramphenicol) were grown in a shaking incubator at 37 °C until reaching an optical density of 0.6–0.7. The temperature was reduced to 16 °C before induction of expression with 1 mM IPTG. Incubation at 16 °C was continued overnight.

The cultures were separated by centrifugation (Beckman J25 with Fiberlite rotor JSP F500, 6000 rpm, 4 °C, 30 min), the supernatant removed and the pellet stored at −80 °C for in excess of 1 h. The PKG's were purified via the histidine tag on HiTrap TALON (cobalt) columns (GE Healthcare) connected to an AKTA-FPLC as per the manufacturer's instructions. Fractions were analysed by sodium dodecyl sulphate–polyacrylamide gel electrophoresis and the main peak concentrated on 10 kDa molecular weight cut-off concentrators (Amicon). Purified proteins were stored in 50% glycerol at −80 °C in single-use aliquots. The final buffer composition of the purified product was: 50 mM Tris/HCl pH 7.5, 0.1 mM EGTA, 150 mM NaCl, 0.1% β-mercaptoethanol, 50% glycerol, 0.03% Brij-35, 1 mM benzamidine and 0.2 mM phenylmethylsulfonyl fluoride.

**Microfluidic assay for recombinant cGMP-dependent protein kinase**. The half-maximal inhibitory concentration (IC$_{50}$) values were determined for test compounds using a microfluidic fluorescent shift assay described. Briefly, compounds were prepared over a 10-well ½ log dilution series in DMSO in duplicate in 50 µL volumes using 384-well polypropylene U-bottomed plates (Thermo Scientific, UK). The plates contained positive/no inhibitor (DMSO only) and negative (no enzyme) controls in columns 1, 2 and 23, 24. The reaction mix for each well consisted of 20 µL of enzyme/peptide mix (1.25 nM PfPKG, 1.5 µM FAM-labelled PKAtide (FAM-GRTGRRNSI-NH2, Cambridge Bioscience, UK) in PfPKG assay buffer (25 mM Hepes (pH 7.4), 20 mM β-glycerophosphate, 2 mM DTT, 10 µM cGMP, 0.01% (w/v) bovine serum albumin (BSA), 0.01% (v/v) Triton X-100)) plus 5 µL of compound. Samples were pre-incubated at room temperature for 30 min and reactions were initiated by addition of 25 µL ATP mix (10 mM MgCl$_2$ and ATP, at Km of enzyme under test (20 µM PfPKG and 90 µM PfPKG$^{T618Q}$), in water). Positive controls were complete reaction mixtures with 10% DMSO and negative controls were reaction mixtures with 10% DMSO but lacking enzyme. Reactions were allowed to proceed for 30 min at room temperature, corresponding to conversion of approximately 10% of the substrate in the DMSO controls. Reactions were terminated by addition of 50 µL stop solution (25 mM EDTA in water). Samples were analysed by electrophoretic separation of substrate and product peaks and fluorescence detection using a Caliper Lab Chip EZ reader (Perkin Elmer, Waltham MA) with 0.2 s sip time, downstream voltage 500 V, upstream voltage 1950 V and pressure 0.5 to 1.5 psi. Substrate and product peak heights were measured and the ratio of the product peak height divided by the sum of the product and substrate peaks were determined using EZ reader software (version 3.0.265.0) to obtain percentage conversion ($P$) values. $P$ values were normalised to percentage activity relative to positive and negative controls were % activity $= 100 \times (P - P_{\text{neg ctrls}})/(P_{\text{pos ctrls}} - P_{\text{neg ctrls}})$ and fitted to obtain IC$_{50}$ values using a 4-parameter logistical fit (XL-fit, IDBS, Guildford UK). Liquid handling stages were conducted on a Biomek robotic liquid handler (Beckman Coulter).

**Immunofluorescence labelling**. Immunofluorescence assays were performed as described in ref. [56]. Briefly, for HA staining, purified cells were fixed with 4% paraformaldehyde and 0.05% glutaraldehyde in PBS for 1 h, permeabilised with 0.1% Triton X-100/PBS for 10 min and blocked with 2% BSA/PBS for 2 h. Primary antibodies were diluted in blocking solution (rat anti-HA clone 3F10, 1:1000 from Sigma-Aldrich, reference 000000011867431001). Anti-rat Alexa488 (Life Technologies, reference A-11006) was used as a secondary antibody together with 4′,6-diamidino-2-phenylindole (DAPI) and diluted 1:1000 in blocking solution. Confocal images were acquired with a LSM800 scanning confocal microscope (Zeiss).

## Data availability

All relevant data are available from the authors on request. Mass spectrometry proteomics data have been deposited to the ProteomeXchange Consortium via the PRIDE partner repository with the dataset identifier PXD011096.

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

## Acknowledgements

We are grateful to David Whalley (LifeArc) for determining the ATP Km of recombinant PKG alleles. We would like to thank Nisha Philip (University of Edinburgh) for sharing the CDPK1-AID-HA line as well as Wesley Van Voorhis and Kayode Ojo (University of Washington) for sharing compound 1294. We thank the excellent service at the proteomics, bioimaging and flow cytometry core facilities at the Faculty of Medicine of the University of Geneva. We also would like to thank Dominique Soldati-Favre for fruitful discussions throughout this study. This work was supported by the Swiss National Science Foundation grant BSSGI0_155852 to M.B. M.B. is an INSERM investigator. P.P. was supported by a SystemsX.ch grant (51TRPO_151032). B.M. is funded by the European Research Council under the European Union's Horizon 2020 Research and Innovation program under grant agreement no. 695596. Work at the Sanger Institute was funded by a core grant from the Wellcome Trust (098051). D.A.B. is funded by a Wellcome grant (106240/Z/14/Z). J.B. is funded by an Investigator Award from the Wellcome (100993/Z/13/Z).

## Author contributions

H.F. A.R.G, N.K., P.P., B.M., E.M.W, Z.A.Z, F.A. and M.B. performed and analysed experiments; J.B., C.D., D.A.B, O.B. and M.B. designed experiments; M.B. wrote the manuscript with help from O.B and A.R.G; all the authors approved the manuscript.

## Additional information

**Competing interests:** The authors declare no competing interests.

