## [Peer Review File · Nature Communications]

Reviewers' comments:

Reviewer #1 (Remarks to the Author):

Review

Fang, Gomes, Klages et al.

In this study, the authors undertake a knockout interaction screen in *P. berghei* to identify novel genetic interactions between phospho-signalling pathways. In so doing, they demonstrate that knockout of CDPK4 in two distinct PKG-hypomorphic backgrounds negatively affects growth, while single mutants are unaffected. The authors proceed to show that this negative epistasis impacts on ring stage formation, suggesting a genetic interaction late in the growth cycle. This finding is further supported by the observation that mutant lines possess a discontinuous IMC, which is postulated to impact negatively on cell rigidity/gliding and, by extension, erythrocyte invasion. Subsequent interactomic analysis further supports a role for CDPK4 in maintaining IMC integrity, as numerous proteins associated with IMC biogenesis/gliding motility are co-immunoprecipitated with this kinase. Intriguingly, the authors find that immunoprecipitation of putative CDPK4 interactors GAP40 (also identified as a putative CDPK4 substrate) and MyoE recovered numerous peptides of SOC6, a protein previously identified in a biochemical screen for CDPK4 substrates. The authors find that deletion of SOC6 closely resembles the aberrant IMC phenotype observed in PKG/CDPK4 mutants.

In addition to these findings, the authors demonstrate that Compound 2, widely used as a PKG-specific inhibitor, has off target effects on CDPK4 function.

Furthermore, by using CDPK4 inhibitors and a line in which CDPK1 can be inducibly degraded, the authors demonstrate that CDPK1 and CDPK4 play complementary roles in ookinete gliding. The authors also postulate, based on use of the abovementioned systems as well as a CDPK3 knockout line, that CDPK3 is involved in the regulation of microneme secretion, while CDPK4 and CDPK1 are not.

On the whole, the data presented is strong and provides valuable novel insights. The chronology in which the data is presented, however, weakens the strength of the paper.

Major comments

- Fig 5 demonstrating that the PKG mutant is a hypomorph should be introduced far earlier (preferably as Fig.2) as the entire premise of a negative epistasis between CDPK4 and PKG relies on this information.

- I would strongly consider a clearer focus on the PKG/CDPK4/IMC story, as there are currently numerous tangents throughout the manuscript that detract from the bigger picture. The Compound 2 data, while certainly important, could have been limited to a single figure (exflagellation data could be moved to supplementary).

Fig 8D – I am not convinced by the microneme secretion data – this experiment should include some form of CDPK3 complementation to confirm that the reduced secretion seen in the KO can be rescued by reintroduction of CDPK3.

Specific comments:

Fig1A – Figure could do with clarification – backgrounds need to be labelled as KO (as PKG and e.g. CDPK1 are currently labelled the same way, while one is tagged, the other is a KO).

Fig2 B,C – Would have been useful, especially in fig 2C to see the single mutants included in the graph. Significance tests should be included

Text states HA epitope tag is used to monitor successful complementation of PKG in PKG T619Q- HA line. This is not possible.

Fig.2G Single mutants should be included, either here, or as 'data not shown'

Fig. 3A Wording in text is incorrect – 'Unexpectedly, in the PfCDPK4-KO line, at 0.5 μ M ring formation was not significantly reduced compared with the 3D7 control while schizont rupture was prevented to the same extent as the WT control' – This should be 'CDPK4 line was reduced significantly less THAN 3D7 control' \diamond this is what the stats are technically showing.

Fig3 B – Text states complementation restored ring formation to WT level - so include a WT control in graph and do the appropriate stats.

In text: Phrasing is incorrect: 'cdpk4s147m did not restore ring formation' – yes it does, but it does not restore the WT phenotype upon c2 treatment – i.e. a reduction in ring formation \ In CDPK4 KO line, absence of CDPK4 is likely compensated by an unknown C2 insensitive kinase. A CDPK4/ CDPK1 KO was generated and shows no growth defect. Would that not suggest that CDPK1 is unlikely a compensating kinase? Or are there several CDPKs that can complement?

Fig4 – Include statistic

Fig 4D – Compound A part of experiment is good but does not get explained in text very well.

Fig8D – ANOVA more appropriate than T test here – this goes for all cases where multiple means are being compared in a single experiment.

Fig8F – Include a blot to confirm CDPK1 degradation in this experiment.

Minor comments & spelling

Fig3 B – middle bar complementation is with CDPK4-myc (not CDPK4).

Fig 4D – Compound A part of experiment is good but should be addressed in the text for clarity.

Fig6B – Clarify in legend that filled circles indicate phosphorylation

Supplementary figure S2 D) complemented shows CDPK4=3xHA, yet there is a PKG gene shown...

No cases of synthetic lethality were observed.

Most notably, no negative interactions among CDPKs was detected...

AKCNOLWEDGEMENTS spelt wrong

To demonstrate that C2 targets CDPK4 during exflagellation \diamond should be 'testing' whether it does this, not 'demonstrating'.

Reviewer #2 (Remarks to the Author):

The study by Fang et al. is an important contribution to our understanding of the signaling pathways that regulate the dynamic processes of invasion and motility in Plasmodium. This work is an extension of several lines of evidence that link cGMP-dependent protein kinase (PKG) and calcium-dependent protein kinases (CDPKs) to the regulation of this process. However, this paper goes a step further in dissecting complex epistatic networks that elucidate redundancies in the pathways. Overall, the work is of superb quality and highly quantitative. My major concerns mainly address the presentation of the data, a few statements that are not fully supported by the observations, and the correct citation of the literature. The article is well written and the concerns raised should be easily addressed.

MAJOR COMMENTS

1. The calculation of the interaction coefficient does not appear to take into account the growth rate defect of both mutations, which would be the appropriate way of looking at synergistic vs. additive effects. Have the authors compared the growth rates of the background clones to determine whether they are really indistinguishable from wild-type? Figure 1B should include the wild-type data for comparison. It may be more informative to plot these interaction coefficients as separate graphs for each background and ordered by kinase. Despite this criticism, the magnitude of the genetic interaction between CDPK4 and PKG is such that it is unlikely to change.
2. The normalization of Figure 3 may be obscuring important data, and the results seem quite strange as presented. Why should removal of the secondary target of C2 remove the block in invasion, when CDPK4 is later associated with invasion (Fig. 3A)? In the CDPK4-KO, C2 results in similar numbers of stalled shizonts, but nearly normal numbers of rings (Fig. 3A); how is this explained? Comparing the results of 3A and 3C it's difficult to understand why the specificity of the compound matters if the secondary target is being removed (CDPK4 is being knocked out. Instead it would seem like C2 is having a secondary effect on a process associated with CDPK4 but not CDPK4 itself.
3. The proposal that "the absence of CDPK4 is likely compensated by an unknown C2-insensitive kinase" is unlikely given that complementation restores the phenotype. Are the authors proposing that in the complemented strains, the presumed compensatory kinase is regulated by the presence of CDPK4?
4. What is the basis for the statement in p.10 "that multiple Ca²⁺-dependent protein kinases might be functionally redundant under physiological activation of PKG."? It is true that CDPK4 is involved in exflagellation, but what is the evidence for the others? If the authors are making this claim on the basis of the literature or the data presented later for merozoites, the statement may be better placed in the discussion.

5. It should be noted that the protein-protein interactions were identified under cross-linking conditions and may not reflect the endogenous protein-protein interactions, but rather proximity between proteins.

6. The IMC phenotype and its association with SOC6 as a target of CDPK4 is rather weak and not necessary for the main thesis of the paper. I would recommend removing these sections given the current length and complexity of the paper. The association with SOC6 is mainly because of IP with CDPK4, which has many caveats. Moreover, the IMC issues could be the result of stalled dying schizonts and not a direct cause of the signaling. Much more work would be needed to bring these observations to the same level as the rest of the paper, and as presented they are simply circumstantial.

7. It is unclear how the authors arrive at the conclusion in p.14 "that another unidentified kinase is involved in this calcium signaling pathway downstream of PKG, CDPK5 being a likely candidate." It seems like this would have to be based on the literature, since there's no evidence presented in Figure 8 for alternative kinases. If this is the case, the statement would be better placed in the discussion.

8. The Toxoplasma literature is relevant to the thesis of the paper and poorly cited. For example, genetic interactions like the ones observed for Plasmodium have previously been reported for Toxoplasma between PKG and TgCDPK3 (the PbCDPK1 ortholog). TgCDPK1 (the PbCDPK4 ortholog) and TgCDPK3 have also been reported to regulate microneme secretion and, in the case of TgCDPK1, in invasion.

MINOR COMMENTS

* additions or modifications displayed in parenthesis.

p.6. "No cases of synthetic lethality (were) observed"

p.6. "...no negative interaction(s) among CDPK(s) were detected..."

Reviewer #3 (Remarks to the Author):

The invasion of host cells by plasmodium parasites involves multiple steps that are regulated by signalling mechanisms. For example, regulated secretion of microneme proteins and phosphorylation of motility related proteins is essential for successful invasion of erythrocytes by plasmodium merozoites. A number of key kinases including calcium dependent protein kinases (CDPKs) and cyclic nucleotide dependent kinases such as cGMP dependent protein kinase G (PKG) have been shown to play a role in these signalling pathways. Here, the authors have used double gene knock outs (KOs) in *P. berghei* and *P. falciparum* to identify kinases and other proteins that interact in signalling events that mediate invasion. The approach is based on the assumption that enhancement of blood stage growth defects in double KOs will identify effectors that interact in signalling pathways related to merozoite invasion. Similar approach is used to study signalling processes that

regulate gametogenesis and exflagellation in sexual stage parasites and motility of ookinetes. A large number of double KOs were studied but only a couple of interacting pairs were identified. The nature of the 'interaction' between the partners in pairs is not always obvious. The following issues need to be addressed:

1. One of the major problems with the study is that it completely ignores the role of cAMP and PKA in signalling events during merozoite invasion. A recent paper published in *Nature Commun.* 2017 Jul 5;8(1):63. doi: 10.1038/s41467-017-00053-1 demonstrated that PfCDPK1 and PfPKA interact with each other to regulate the invasion process. Was PfPKA included in the study to look for interacting partners? How do the authors defend their final model (Figure 9) which only includes PKG and does not consider potential interactions with PKA and role of cAMP.

2. The authors make the assumption that phosphodiesterase inhibitors such as zaprinast solely raise cGMP levels. In fact, zaprinast can raise cAMP levels as well. The authors should measure cGMP or cAMP levels in merozoites to confirm that zaprinast treatment only raises cGMP levels.

3. The individual strains PbCDPK4-KO, PbPKGT619Q-HA and PbPKG-HA do not have a blood stage growth defect when compared to wild type (WT) *P. berghei* strain. However, both double mutants PbCDPK4-KO - PbPKGT619Q-HA and PbCDPK4-KO - PbPKG-HA had significant growth defects. Complementation of double mutants with non-epitope tagged PbCDPK4 or PbPKG restored normal growth. The authors later demonstrate that the gatekeeper mutation T619Q affects its affinity for ATP and reduces calcium mobilisation following PKG activation. This may explain the phenotype of the PbCDPK4-KO - PbPKGT619Q-HA double mutant. However, what about the PbCDPK4-KO - PbPKG-HA double mutant? Why is invasion impaired in this double mutant as well? Does the HA tag or 3'UTR in the transgenic parasites affect PbPKG expression leading to the effects observed? The same concerns are relevant for CDPK4-HA tagging. Why is complementation with non-tagged CDPK4 more effective at restoration of normal levels compared to HA-tagged CDPK4? Is expression of CDPK4 affected by HA tagging or the 3'UTR introduced in the transgenic parasite? The authors refer to these mutations being 'hypomorphic'. Is there any change in CDPK4 protein levels or specific activity? Without addressing these issues, the conclusions about nature of interactions between PfCDPK4 and PfPKG remain speculative and are not conclusive.

4. The authors have only studied asexual blood stage parasite growth both in case of *P. berghei* and *P. falciparum*. They have not analysed specific events such as egress, invasion or microneme secretion. In some cases, they have scored 'ring formation'. However, even this involves egress and invasion. How do the authors attribute role of CDPKs to specific events such as microneme secretion or motility? The model presented in Figure 9 is thus highly speculative.

5. On Page 8, authors report that replacement of S448 and S449 with alanine in GAP40 does not affect blood stage parasite growth. The authors interpret this to suggest that there is 'plasticity' in components used to assemble the glideosome i.e. GAP40 S448/S449

phosphorylation is important but can be replaced functionally by some other protein in absence of phosphorylation. Instead a simpler explanation may be that S448/S449 phosphorylation on GAP40 is just not functionally important. Why have the authors not considered such an explanation for their observaiton?

Minor points

1. In experiments reported in Figure 3, why is PbCDPK4 used to complement PfCDPK4-KO instead of PfCDPK4?
2. Page 8, line 8 – mention 0.5 μ M of C2
3. Is the difference in ε values for PbCDPK4-KO - PbPKGT619Q-HA and PbCDPK4-KO - PbPKG-HA statistically significant?
4. The authors observe defects in the inner membrane complex (IMC) of the merozoites of the PbCDPK4-KO - PbPKGT619Q-HA double mutants compared to WT strain. The authors suggest that this may lead to the defect in erythrocyte invasion observed in case of the double mutants. The authors also suggest that the IMC defect may lead to 'weaker cell rigidity and defective gliding'. This statement is purely speculative with no data provided to substantiate it.
5. Page 8 – the observation that C2 inhibits PfCDPK4 as well as PKG is referred to as synergistic effect. Why is it synergistic? It seems more like an off-target effect that complicates interpretation of the results. The nomenclature for the type of effect should be changed.
6. The observation that C2 has off-target effects and inhibits PfCDPK4 in addition to PfPKG is important because previous studies have assumed that C2 specifically inhibits PfPKG. How does this affect current signalling models related to merozoite invasion?

Reviewer #1 (Remarks to the Author):

Review

Fang, Gomes, Klages et al.

In this study, the authors undertake a knockout interaction screen in *P. berghei* to identify novel genetic interactions between phospho-signalling pathways. In so doing, they demonstrate that knockout of CDPK4 in two distinct PKG-hypomorphic backgrounds negatively affects growth, while single mutants are unaffected. The authors proceed to show that this negative epistasis impacts on ring stage formation, suggesting a genetic interaction late in the growth cycle. This finding is further supported by the observation that mutant lines possess a discontinuous IMC, which is postulated to impact negatively on cell rigidity/gliding and, by extension, erythrocyte invasion. Subsequent interactomic analysis further supports a role for CDPK4 in maintaining IMC integrity, as numerous proteins associated with IMC biogenesis/gliding motility are co-immunoprecipitated with this kinase. Intriguingly, the authors find that immunoprecipitation of putative CDPK4 interactors GAP40 (also identified as a putative CDPK4 substrate) and MyoE recovered numerous peptides of SOC6, a protein previously identified in a biochemical screen for CDPK4 substrates. The authors find that deletion of SOC6 closely resembles the aberrant IMC phenotype observed in PKG/CDPK4 mutants. In addition to these findings, the authors demonstrate that Compound 2, widely used as a PKG-specific inhibitor, has off target effects on CDPK4 function. Furthermore, by using CDPK4 inhibitors and a line in which CDPK1 can be inducibly degraded, the authors demonstrate that CDPK1 and CDPK4 play complementary roles in ookinete gliding. The authors also postulate, based on use of the abovementioned systems as well as a CDPK3 knockout line, that CDPK3 is involved in the regulation of microneme secretion, while CDPK4 and CDPK1 are not.

On the whole, the data presented is strong and provides valuable novel insights. The chronology in which the data is presented, however, weakens the strength of the paper.

We agree with the reviewer that the manuscript was rather dense and thank them for their recommendations to ease the understanding of the results. We have included new experiments to confirm the role of CDPK3 in ookinete microneme secretion.

Major comments

- Fig 5 demonstrating that the PKG mutant is a hypomorph should be introduced far earlier (preferably as Fig.2) as the entire premise of a negative epistasis between CDPK4 and PKG relies on this information.

We have significantly changed the manuscript organisation, as advised by the reviewer, and now describe the effects of the PKG mutation in Fig. 2.

- I would strongly consider a clearer focus on the PKG/CDPK4/IMC story, as there are currently numerous tangents throughout the manuscript that detract from the bigger picture. The Compound 2 data, while certainly important, could have been limited to a single figure (exflagellation data could be moved to supplementary).

We have moved the exflagellation data (text and figures) to the supplementary information and show the compound 2 data in Fig. 2. For clarity and space constraints, we have also removed the

results describing the effect of CDPK4 myristoylation in asexual stages. These results are intriguing but do not contribute to a better understanding of CDPK4 functions in asexual stages.

Fig 8D – I am not convinced by the microneme secretion data – this experiment should include some form of CDPK3 complementation to confirm that the reduced secretion seen in the KO can be rescued by reintroduction of CDPK3.

To ascertain that the effect on secretion was not due to an abnormal development of CDPK3-KO ookinetes, we have first confirmed that the CDPK3-KO/CeLTOS-3xHA transgenics shows the same ookinete conversion rate as the CeLTOS-3xHA control. This data has been added in Supplementary Fig. 9C. The gliding motility phenotype associated with *cdpk3* deletion was previously documented in three independent reports (PMIDs 16796674, 16430692, and 24594931), one of which (16796674) showing successful complementation with an episomally maintained *cdpk3* gene that restored mosquito transmission to wild-type levels. Despite this strong line of evidence for a role of CDPK3 in ookinete motility, we have first recycled the hDHFR selection marker in the CDPK3-KO/CeLTOS-3xHA line and cis-complemented the resulting clone with a CDPK3-3xHA allele or trans-complemented the transgenics using an artificial chromosome containing a genomic insert including *cdpk3* and its 5' and 3' UTRs. Both approaches restored CeLTOS-3xHA secretion to wild type levels. This data has been added in Supplementary Fig. 9D and E.

Specific comments:

Fig1A – Figure could do with clarification – backgrounds need to be labelled as KO (as PKG and e.g. CDPK1 are currently labelled the same way, while one is tagged, the other is a KO).

We thank the reviewer for this comment and have amended the figure accordingly.

Fig2 B,C – Would have been useful, especially in fig 2C to see the single mutants included in the graph. Significance tests should be included

Text states HA epitope tag is used to monitor successful complementation of PKG in PKG T619Q- HA line. This is not possible.

We have tried to include all the growth curves on the same chart but the result was confusing as the control growths were hardly distinguishable. We are presenting the data next to each other and have included significance tests confirming the increased growth rates upon complementation albeit reduced compared with the wild type control.

We did complement the PKG^{T619Q}-3xHA line with a PKG-3xHA allele by transfecting a PKG-3xHA targeting construct. We confirmed the gatekeeper substitution in the resulting clone by Sanger sequencing. The sequence traces have now been included in the genotyping figure (Supplementary Fig. 2F). We however did not use the 3xHA tag as a marker of complementation as it would not have allowed to discriminate between the two alleles.

Fig.2G Single mutants should be included, either here, or as 'data not shown'

We have now included the data for single mutants in Fig. 1G.

Fig. 3A Wording in text is incorrect – ‘Unexpectedly, in the PfCDPK4-KO line, at 0.5 μ M ring formation was not significantly reduced compared with the 3D7 control while schizont rupture was prevented to the same extent as the WT control’ – This should be ‘CDPK4 line was reduced significantly less THAN 3D7 control’ \diamond this is what the stats are technically showing.

It appears the normalisation we used to present the results was confusing and not showing the whole range of observed variations. We are now showing the full EC₅₀ curves for both schizont rupture and ring formation showing that the CDPK4-KO line is more resistant than the WT control for ring formation at a C2 concentration of 0.5 μ M. We have also amended the text to better describe the results.

Fig3 B – Text states complementation restored ring formation to WT level - so include a WT control in graph and do the appropriate stats.

In text: Phrasing is incorrect: ‘cdpk4s147m did not restore ring formation’ – yes it does, but it does not restore the WT phenotype upon c2 treatment – i.e. a reduction in ring formation

As for Fig. 3A, we are now showing the full EC₅₀ curves for both schizont rupture and ring formation showing that complementation the CDPK4-KO line with a PbCDPK4-2xmyc allele lowers ring formation at a C2 concentration of 0.5 μ M while complementation with a PbCDPK4^{S147M}-2xmyc allele does not. We have also amended the text to better describe the results.

In CDPK4 KO line, absence of CDPK4 is likely compensated by an unknown C2 insensitive kinase. A CDPK4/ CDPK1 KO was generated and shows no growth defect. Would that not suggest that CDPK1 is unlikely a compensating kinase? Or are there several CDPKs that can complement?

We agree that the absence of growth defect in the CDPK4-KO/CDPK1-KO line argues against a compensation by CDPK1. However we were not able to generate a PKG^{T619Q}-3xHA/CDPK4-KO/CDPK1-KO line and combined depletion of CDPK1 and inhibition of CDPK4 in ookinete led to a reduction of gliding speed. This suggests that CDPK1 may be complementing CDPK4 but that an unidentified kinase is likely involved. We have amended the discussion to better explain this point and proposed that CDPK5 is a likely candidate without excluding the possibility that other kinases including PKA might be involved in this signalling network.

Fig4 – Include statistic

We have included statistics for this figure that is now shown in Supplementary Fig. 4.

Fig 4D – Compound A part of experiment is good but does not get explained in text very well.

We have rephrased the text to better explain the rationale behind the Compound A experiment.

Fig8D – ANOVA more appropriate than T test here – this goes for all cases where multiple means are being compared in a single experiment.

We thank the reviewer for this judicious advice. We now use ANOVA for multiple comparisons.

Fig8F – Include a blot to confirm CDPK1 degradation in this experiment.

The blot is now shown in the source files.

Minor comments & spelling

Fig3 B – middle bar complementation is with CDPK4-myc (not CDPK4).

We have amended the description of the parasite lines.

Fig 4D – Compound A part of experiment is good but should be addressed in the text for clarity.

We have rephrased the text to better explain the rationale behind the Compound A experiment.

Fig6B – Clarify in legend that filled circles indicate phosphorylation

We now specify in the legend that filled circles indicate phosphorylation.

Supplementary figure S2 D) complemented shows CDPK4-3xHA, yet there is a PKG gene shown...

We show genotyping data for both *pkg* and *cdpk4* loci to confirm the complementation of *cdpk4* in the *pkg*^{T619Q}-3xHA background.

No cases of synthetic lethality were observed.

We have changed the text accordingly.

Most notably, no negative interactions among CDPKs was detected...

We have changed the text accordingly.

AKCNOLWEDGEMENTS spelt wrong

We have corrected the typo.

To demonstrate that C2 targets CDPK4 during exflagellation ◊ should be 'testing' whether it does this, not 'demonstrating'.

We have changed the text accordingly.

Reviewer #2 (Remarks to the Author):

The study by Fang et al. is an important contribution to our understanding of the signaling pathways that regulate the dynamic processes of invasion and motility in Plasmodium. This work is an extension of several lines of evidence that link cGMP-dependent protein kinase (PKG) and calcium-dependent protein kinases (CDPKs) to the regulation of this process. However, this paper goes a step further in dissecting complex epistatic networks that elucidate redundancies in the pathways. Overall, the work is of superb quality and highly quantitative. My major concerns mainly address the presentation of the data, a few statements that are not fully supported by the observations, and the correct citation of the literature. The article is well written and the concerns raised should be easily addressed.

We thank reviewer 2 for their very supportive and constructive comments. We have extensively modified the presentation of the data, toned down some statements when required, and added new references.

MAJOR COMMENTS

1. The calculation of the interaction coefficient does not appear to take into account the growth rate defect of both mutations, which would be the appropriate way of looking at synergistic vs. additive effects. Have the authors compared the growth rates of the background clones to determine whether they are really indistinguishable from wild-type? Figure 1B should include the wild-type data for comparison. It may be more informative to plot these interaction coefficients as separate graphs for each background and ordered by kinase. Despite this criticism, the magnitude of the genetic interaction between CDPK4 and PKG is such that it is unlikely to change.

As suggested by reviewer 2 we now show a plot of the fitness scores as separate graphs for each background, including, the WT and ordered by kinase.

2. The normalization of Figure 3 may be obscuring important data, and the results seem quite strange as presented. Why should removal of the secondary target of C2 remove the block in invasion, when CDPK4 is later associated with invasion (Fig. 3A)? In the CDPK4-KO, C2 results in similar numbers of stalled schizonts, but nearly normal numbers of rings (Fig. 3A); how is this explained? Comparing the results of 3A and 3C it's difficult to understand why the specificity of the compound matters if the secondary target is being removed (CDPK4 is being knocked out. Instead it would seem like C2 is having a secondary effect on a process associated with CDPK4 but not CDPK4 itself.

It appears the normalisation we used to present the results was confusing. We are now showing the full EC₅₀ curves for both schizont rupture and ring formation showing that the CDPK4-KO line is more resistant than the WT control for ring formation at a C2 concentration of 0.5 μM. We have also amended the text to better describe the results.

3. The proposal that “the absence of CDPK4 is likely compensated by an unknown C2-insensitive kinase” is unlikely given that complementation restores the phenotype. Are the authors proposing

that in the complemented strains, the presumed compensatory kinase is regulated by the presence of CDPK4?

We think the way we presented results confused the reviewer and have changed the representation of the data.

4. What is the basis for the statement in p.10 “that multiple Ca²⁺-dependent protein kinases might be functionally redundant under physiological activation of PKG.”? It is true that CDPK4 is involved in exflagellation, but what is the evidence for the others? If the authors are making this claim on the basis of the literature or the data presented later for merozoites, the statement may be better placed in the discussion.

We concur with the reviewer and have kept this statement for the discussion.

5. It should be noted that the protein-protein interactions were identified under cross-linking conditions and may not reflect the endogenous protein-protein interactions, but rather proximity between proteins.

We agree with the reviewer and we have been very careful not to mention the terms “protein-protein interaction” or “protein complex” but rather co-immunoprecipitation following cross-linking. We now indicate cross-linking in the result section and not only in the method section.

6. The IMC phenotype and its association with SOC6 as a target of CDPK4 is rather weak and not necessary for the main thesis of the paper. I would recommend removing these sections given the current length and complexity of the paper. The association with SOC6 is mainly because of IP with CDPK4, which has many caveats. Moreover, the IMC issues could be the result of stalled dying schizonts and not a direct cause of the signaling. Much more work would be needed to bring these observations to the same level as the rest of the paper, and as presented they are simply circumstantial.

We fully concur with reviewer 2 and we now clearly state that we have not investigated how SOC6 phosphorylation affects its function. Investigating both the molecular function of SOC6 and the role of its phosphorylation would represent a full project in itself. We however felt it was important to confirm that this newly identified substrate of CDPK4 is important for IMC stability or formation.

7. It is unclear how the authors arrive at the conclusion in p.14 “that another unidentified kinase is involved in this calcium signaling pathway downstream of PKG, CDPK5 being a likely candidate.” It seems like this would have to be based on the literature, since there’s no evidence presented in Figure 8 for alternative kinases. If this is the case, the statement would be better placed in the discussion.

We have moved this statement to the discussion and better explain why CDPK5 represents an interesting candidate to be studied further.

8. The *Toxoplasma* literature is relevant to the thesis of the paper and poorly cited. For example, genetic interactions like the ones observed for Plasmodium have previously been reported for Toxoplasma between PKG and TgCDPK3 (the PbCDPK1 ortholog). TgCDPK1 (the PbCDPK4 ortholog) and TgCDPK3 have also been reported to regulate microneme secretion and, in the case of TgCDPK1, in invasion.

Reviewer 2 is absolutely right. We have now added *Toxoplasma* references describing the interplay between Ca^{2+} , cGMP and cAMP, the role of CDPKs during egress and invasion and the redundancy of CDPKs.

MINOR COMMENTS

* additions or modifications displayed in parenthesis.

p.6. "No cases of synthetic lethality (were) observed"
We have changed the text accordingly.

p.6. "...no negative interaction(s) among CDPK(s) were detected..."
We have changed the text accordingly.

Reviewer #3 (Remarks to the Author):

The invasion of host cells by plasmodium parasites involves multiple steps that are regulated by signalling mechanisms. For example, regulated secretion of microneme proteins and phosphorylation of motility related proteins is essential for successful invasion of erythrocytes by plasmodium merozoites. A number of key kinases including calcium dependent protein kinases (CDPKs) and cyclic nucleotide dependent kinases such as cGMP dependent protein kinase G (PKG) have been shown to play a role in these signalling pathways. Here, the authors have used double gene knock outs (KOs) in *P. berghei* and *P. falciparum* to identify kinases and other proteins that interact in signalling events that mediate invasion. The approach is based on the assumption that enhancement of blood stage growth defects in double KOs will identify effectors that interact in signalling pathways related to merozoite invasion. Similar approach is used to study signalling processes that regulate gametogenesis and exflagellation in sexual stage parasites and motility of ookinetes. A large number of double KOs were studied but only a couple of interacting pairs were identified. The nature of the 'interaction' between the partners in pairs is not always obvious. The following issues need to be addressed:

We thank reviewer 3 for their constructive comments. We have clarified some points, toned down statements when speculative and included more literature to discuss the potential role of other kinases in the signalling network we describe in this study.

1. One of the major problems with the study is that it completely ignores the role of cAMP and PKA in signalling events during merozoite invasion. A recent paper published in Nature Commun. 2017 Jul 5;8(1):63. doi: 10.1038/s41467-017-00053-1 demonstrated that PfCDPK1 and PfPKA interact with each other to regulate the invasion process. Was PfPKA included in the study to look for interacting partners? How do the authors defend their final model (Figure 9) which only includes PKG and does not consider potential interactions with PKA and role of cAMP.

We fully agree with reviewer 3 that PKA is the elephant in the room. cAMP signalling and PKA have indeed both been shown to be involved in the process of invasion although the exact molecular roles of this signalling pathway remain unclear. This is probably best explained by the absence of robust tools to study cAMP pathways and the intricate nature of signalling networks during such a very short time window.

The nature of our genetic screen did not allow to investigate negative interactions with essential kinases such as PKA as, so far, we do not have a robust conditional system allowing to systematically screen for interactions with or among essential kinases. Similarly, the interactomic analysis did not detect peptides for PKA. This does not exclude a functional relationship between PKA with cGMP or calcium signalling.

We are now stating in the discussion that multiple other kinases may be part of the network, including PKA. However, as our study did not find positive evidence for a functional relationship with PKA or other essential kinases, we decided to only show the relationships we have identified in this study. We think that a more comprehensive model would require an extensive citation of the literature that would be more appropriate in a dedicated review. In our final model, we now clearly state that the proposed model is deduced from this work and that other signalling components may be part of the network.

2. The authors make the assumption that phosphodiesterase inhibitors such as zaprinast solely raise cGMP levels. In fact, zaprinast can raise cAMP levels as well. The authors should measure cGMP or cAMP levels in merozoites to confirm that zaprinast treatment only raises cGMP levels.

Once again we fully concur with reviewer 3. It was indeed previously shown that zaprinast raises both cGMP and cAMP levels in erythrocytes infected with *P. falciparum* asexual stages (PMID 18590734) and that BIPPO triggers PKA-dependent processes in merozoites (PMID 25555060). We have previously shown that the calcium response and egress of merozoite induced by zaprinast could be completely reverted by the specific inhibition of PKG (PMID 24594931) suggesting that the effect on calcium signalling is mainly dependent on PKG. Even though cAMP levels are possibly raised under our experimental conditions, this would not affect the conclusion of this experiment as, it was designed to investigate whether a physiological difference between the wild type and PKG^{T618Q} isoforms could be observed in response to zaprinast or BIPPO using calcium as a readout. This highlighted that PKG^{T618Q} is less responsive to zaprinast or BIPPO suggesting it is defective to mediate a calcium response irrespective of the specificity of zaprinast and BIPPO.

3. The individual strains PbCDPK4-KO, PbPKGT619Q-HA and PbPKG-HA do not have a blood stage growth defect when compared to wild type (WT) *P. berghei* strain. However, both double mutants PbCDPK4-KO - PbPKGT619Q-HA and PbCDPK4-KO - PbPKG-HA had significant growth defects. Complementation of double mutants with non-epitope tagged PbCDPK4 or PbPKG restored normal growth. The authors later demonstrate that the gatekeeper mutation T619Q affects its affinity for ATP and reduces calcium mobilisation following PKG activation. This may explain the phenotype of the PbCDPK4-KO - PbPKGT619Q-HA double mutant. However, what about the PbCDPK4-KO - PbPKG-HA double mutant? Why is invasion impaired in this double mutant as well? Does the HA tag or 3'UTR in the transgenic parasites affect PbPKG expression leading to the effects observed? The same concerns are relevant for CDPK4-HA tagging. Why is complementation with non-tagged CDPK4 more effective at restoration of normal levels compared to HA-tagged CDPK4? Is expression of CDPK4 affected by HA tagging or the 3'UTR introduced in the transgenic parasite? The authors refer to these mutations being 'hypomorphic'. Is there any change in CDPK4 protein levels or specific activity? Without addressing these issues, the conclusions about nature of interactions between PfCDPK4 and PfPKG remain speculative and are not conclusive.

Our data clearly indicates 3xHA tag or the generic 3'UTR subtly affect the function of PKG or CDPK4. Unfortunately, we currently do not know how these genetic modifications may affect the function of these kinases. These two factors may affect various parameters including protein expression or stability, protein localisation, substrate specificity, protein-protein interaction or kinase activity. It is important to note that for most of the phenotypes of single mutants investigated so far, both PKG-HA and CDPK4-3xHA lines behaved as the wild type. The only stage-specific difference observed was a decreased exflagellation rate in the CDPK4-3xHA line. Interestingly, a line expressing a CDPK4-2xmyc allele under the control of the same 3'UTR did not show any defect in exflagellation nor a difference in protein expression or localisation (PMID 28481199) suggesting that the HA tag itself may affect the kinase function without obviously affecting the protein expression nor localisation. However complementation with non-tagged alleles of PKG and CDPK4 did not fully restore growth to wild levels indicating that the generic 3'UTR used is also imposing a fitness cost. In the absence of specific antibodies, we are currently not able to test for a potential difference in protein localisation or expression.

Importantly, we here provide strong evidence that the gatekeeper mutation induces a fitness cost on the kinase function as i) this mutation significantly exacerbates the fitness cost of deleting *cdpk4*

in the PKG-3xHA background in *P. berghei* asexual blood stages as shown by the cis-complementation of PKG^{T619Q}-HA/CDPK4-KO line with a PKG-HA allele, ii) this mutation lowers PKG-dependent calcium signals in *P. falciparum* schizonts, iii) this mutation affects affinity for ATP in a recombinant PKG enzyme. This is in addition to a previous work (PMID27425827) showing that *P. berghei* sporozoites expressing the PKG^{T619Q}-3xHA allele are less invasive than sporozoites expressing the PKG-3xHA allele.

We have now included a statement on the possible reasons on how the 3xHA tag or the generic 3'UTR could affect the function of PKG or CDPK4.

4. The authors have only studied asexual blood stage parasite growth both in case of *P. berghei* and *P. falciparum*. They have not analysed specific events such as egress, invasion or microneme secretion. In some cases, they have scored 'ring formation'. However, even this involves egress and invasion. How do the authors attribute role of CDPKs to specific events such as microneme secretion or motility? The model presented in Figure 9 is thus highly speculative.

Our flow cytometry assay allows to score egress and ring formation as shown in figure S5 but this may have been not clear due to the data normalisation we used, as also raised by reviewers 1 and 2. The assay showed that for the same number of ruptured schizonts, the CDPK4-KO line was giving rise to more ring parasites than the wild type. This suggested that in the wild type, C2 has a second target important for invasion that is absent or not active in the CDPK4-KO line. We are now showing the full EC₅₀ curves for both schizont rupture and ring formation. We have also amended the text to better describe the results.

As indicated by the reviewer, we did not investigate microneme secretion in asexual blood stages and we cannot exclude a role of CDPK4 in microneme secretion as CDPK1 was previously shown to be involved in microneme secretion (PMID28680058). We now mention this possibility in the discussion.

We initially included CDPK5 in our model as we were extremely enthusiastic about the then just published results by the Dvorin group showing that CDPK5 requirement for microneme secretion could be bypassed by over-activation of PKG (PMID 29487234). As reviewer 3 indicates, the link with our work is speculative and we have decided to remove CDPK5 from our model to only show the results obtained in this study.

5. On Page 8, authors report that replacement of S448 and S449 with alanine in GAP40 does not affect blood stage parasite growth. The authors interpret this to suggest that there is 'plasticity' in components used to assemble the glideosome i.e. GAP40 S448/S449 phosphorylation is important but can be replaced functionally by some other protein in absence of phosphorylation. Instead a simpler explanation may be that S448/S449 phosphorylation on GAP40 is just not functionally important. Why have the authors not considered such an explanation for their observation?

We thank the reviewer for this observation and we have included this evident hypothesis in the result section.

Minor points

1. In experiments reported in Figure 3, why is PbCDPK4 used to complement PfCDPK4-KO instead of PfCDPK4?

We initiated cloning of PfCDPK4 to complement the PfCDPK4-KO however this proved difficult. As we had the PbCDPK4-2xmyc and PbCDPK4^{S147M}-2xmyc constructs available from a previous study, we transfected them while cloning the PfCDPK4 corresponding constructs. As the PbCDPK4-2xmyc construct partially restored the PfCDPK4-KO phenotype and the results in gametogenesis or with compound A both indicated that C2 had off-target through CDPK4 inhibition, we considered the evidence that C2 also targeted CDPK4 asexual stages was strong enough. We clearly indicate in the figure legend that the complementation was performed with *P. berghei* alleles.

2. Page 8, line 8 – mention 0.5 μ M of C2

We have significantly modified the text and we now pay attention to mention the compound concentration when necessary.

3. Is the difference in ϵ values for PbCDPK4-KO - PbPKGT619Q-HA and PbCDPK4-KO - PbPKG-HA statistically significant?

The experimental settings used for the signature tagged mutagenesis approach did not give us enough statistical power to ascertain a significant difference in the ϵ value between the PbCDPK4-KO/PbPKG^{T619Q}-HA and PbCDPK4-KO/PbPKG-HA pairs. However, complementation of the PKG^{T619Q}-HA/CDPK4-KO with a PKG-HA allele leading to a PKG-HA/CDPK4-KO line is associated with a faster growth rate compared with the PbCDPK4-KO/PbPKG^{T619Q}-HA line. This indicates that a PKG-HA/CDPK4-KO line has a significantly faster growth rate than a PKG^{T619Q}-HA/CDPK4-KO line and that the gatekeeper mutation exacerbates the fitness cost imposed by the 3xHA tag and the generic 3'UTR.

4. The authors observe defects in the inner membrane complex (IMC) of the merozoites of the PbCDPK4-KO - PbPKGT619Q-HA double mutants compared to WT strain. The authors suggest that this may lead to the defect in erythrocyte invasion observed in case of the double mutants. The authors also suggest that the IMC defect may lead to 'weaker cell rigidity and defective gliding'. This statement is purely speculative with no data provided to substantiate it.

We fully concur with reviewer 3 that our observation is correlative and the effects observed for ring formation may be also caused by the deregulation of other cellular processes our approaches did not allow to detect. We have removed this statement from the results and only mention the correlation in the discussion.

5. Page 8 – the observation that C2 inhibits PfCDPK4 as well as PKG is referred to as synergistic effect. Why is it synergistic? It seems more like an off-target effect that complicates interpretation of the results. The nomenclature for the type of effect should be changed.

We have been tempted to speculate that targeting both CDPK4 and PKG by C2 is producing a synergistic effect as C2 is silent in the PKG^{T618Q} line. However, we fully agree with reviewer 3's

comment that we do not demonstrate synergism in this case and we are now using the term dual targeting.

6. The observation that C2 has off-target effects and inhibits PfCDPK4 in addition to PfPKG is important because previous studies have assumed that C2 specifically inhibits PfPKG. How does this affect current signalling models related to merozoite invasion?

The selectivity of most pharmacological agents is often limited and difficult to assess. In the case of C2, the observed shifts in EC_{50} upon treatment of lines expressing C2-resistant PKG alleles ensured that the cellular and molecular phenotypes observed for egress, rounding-up, ookinete motility, sporozoite motility and calcium mobilisation were mainly caused by the inhibition of PKG. However, depending on the stages investigated, EC_{50} values only shifted between 1.5- to 50-fold and so it is clear that there are additional targets depending on the concentration used and the stage investigated.

In this work, we show that C2 also targets CDPK4 during invasion but as, in this case, CDPK4 is functionally redundant and downstream of PKG leading to subtle effect in the WT background and non-detectable effects in the line expressing the PKG^{T618Q} allele. During gametogenesis, CDPK4 is essential and the off-target effect of C2 is obvious.

Altogether we do not think that this dual specificity affects our current knowledge regarding the role of PKG in merozoite egress, gametogenesis, ookinete motility and sporozoite motility when the results were carefully confirmed with lines expressing a C2-resistant allele. However, we think this may affect results obtained when mutating other enzymes in the pathway such as CDPK1 or possibly CDPK5. Such changes may either affect the sensitivity of the mutated kinase to C2 or affect the kinase activity that would lead to stronger requirements for CDPK4 or PKG. In this cases, it would be necessary to ascertain the targets of C2 and the effect of the genetic modifications before drawing any conclusion.

REVIEWERS' COMMENTS:

Reviewer #1 (Remarks to the Author):

Most of my concerns have been addressed by the authors and I am happy with the manuscript as it stands.

Reviewer #2 (Remarks to the Author):

The authors have worked to address most of the concerns raised in my previous review. Given the complexity of the manuscript, I am satisfied with their responses, and understand that many of the issues raised may not be addressed in a single study. Although, I don't believe the point warrants further revisions, I would argue that the difference in C2 EC50s observed in WT compared to the CDPK4 KO may not be the result of "a second target important for invasion that is absent or not active in the CDPK4-KO line," but more likely caused by an upregulation of PKG in the CDPK1-KO line—as perhaps expected from the epistasis they document.

Reviewer #3 (Remarks to the Author):

The authors have addressed most of the comments that were raised by Reviewer 3. The authors should address the following two comments.

1. It was suggested that the authors should not ignore the potential role of cAMP and PKA in signalling events during merozoite invasion. While the authors do now mention the potential role of PKA in signalling during invasion by apicomplexan parasites in the Discussion, in the Introduction they still focus only on cGMP and calcium. The authors should mention the potential roles of cAMP and PKA in regulating invasion related events in the Introduction as well.
2. Page 7 – The ultrastructural analysis by the TEM refers to Figures 2G and H instead of Figures 1G and H.

REVIEWERS' COMMENTS:

Reviewer #1 (Remarks to the Author):

Most of my concerns have been addressed by the authors and I am happy with the manuscript as it stands.

Reviewer #2 (Remarks to the Author):

The authors have worked to address most of the concerns raised in my previous review. Given the complexity of the manuscript, I am satisfied with their responses, and understand that many of the issues raised may not be addressed in a single study. Although, I don't believe the point warrants further revisions, I would argue that the difference in C2 EC50s observed in WT compared to the CDPK4 KO may not be the result of "a second target important for invasion that is absent or not active in the CDPK4-KO line," but more likely caused by an upregulation of PKG in the CDPK1-KO line—as perhaps expected from the epistasis they document.

We agree that up-regulation of PKG activity is a possible alternative. However the results obtained in this study favour a second off-target, as we demonstrate that i) no difference in basal calcium levels can be detected in the CDPK4-KO line, ii) C2 targets both PKG and CDPK4, iii) that a CDPK4-S147M allele does not complement CDPK4 deletion in presence of C2 both in *P. berghei* gametocytes and *P. falciparum* schizonts, and, most importantly, iv) that Compound A, which targets PKG but not CDPK4, shows the same EC50 in both WT and CDPK4-KO lines.

Reviewer #3 (Remarks to the Author):

The authors have addressed most of the comments that were raised by Reviewer 3. The authors should address the following two comments.

1. It was suggested that the authors should not ignore the potential role of cAMP and PKA in signalling events during merozoite invasion. While the authors do now mention the potential role of PKA in signalling during invasion by apicomplexan parasites in the Discussion, in the Introduction they still focus only on cGMP and calcium. The authors should mention the potential roles of cAMP and PKA in regulating invasion related events in the Introduction as well.

We now mention in the introduction that egress and invasion rely on multiple intracellular messengers including cGMP and calcium.

2. Page 7 - The ultrastructural analysis by the TEM refers to Figures 2G and H instead of Figures 1G and H.

We thank reviewer 3 for spotting this typo. We have corrected it.